# Fault Diagnosis of Rolling Bearings Based on Improved Kurtogram in Varying Speed Conditions

**Yong Ren** **, Wei Li \*, Bo Zhang** , **Zhencai Zhu and Fang Jiang**

School of Mechanical and Electrical Engineering, China University of Mining and Technology, Xuzhou 221116, China; reny_cumt@163.com (Y.R.); zbcumt@163.com (B.Z.); zzc_cmee@163.com (Z.Z.); jiangfan25709@163.com (F.J.)
\* Correspondence: liwei_cmee@163.com

**Abstract:** Envelope analysis is a widely used method in fault diagnoses of rolling bearings. An optimal narrowband chosen for the envelope demodulation is critical to obtain high detection accuracy. To select the narrowband, the fast kurtogram (FK), which computes the kurtosis of a set of filtered signals, is introduced to detect cyclic transients in a signal, and the zone with the maximum kurtosis is the optimal frequency band. However, the kurtosis value is affected by rotating frequencies and is sensitive to large random impulses which normally occur in industrial applications. These factors weaken the performance of the FK for extracting weak fault features. To overcome these limitations, a novel feature named Order Spectrum Correlated Kurtosis (OSCK) is proposed, replacing the kurtosis index in the FK, to construct an improved kurtogram called Fast Order Spectrum Correlated Kurtogram (FOSCK). A band-pass filter is used to extract the optimal frequency band signal corresponding to the maximum OSCK. The envelope of the filtered signal is calculated using the Hilbert transform, and a low-pass filter is employed to eliminate the trend terms of the envelope. Then, the non-stationary filtered envelope is converted in the time domain into the stationary envelope in the angular domain via Computed Order Tracking (COT) to remove the effects of the speed fluctuation. The order structure of the angular domain envelope signal can then be used to determine the type of fault by identifying its characteristic order. This method offers several merits, such as fine order spectrum resolution and robustness to both random shock and heavy noise. Additionally, it can accurately locate the bearing fault resonance band within a relatively large speed fluctuation. The effectiveness of the proposed method is verified by a number of simulations and experimental bearing fault signals. The results are compared with several existing methods; the proposed method outperforms others in accurate bearing fault feature extraction under varying speed conditions.

**Keywords:** fault diagnosis; fast kurtogram; order spectrum correlated kurtosis; rolling bearing; non-stationary

## 1. Introduction

Rolling bearings are among the most commonly used support elements in rotating machinery. They are prone to faults under harsh working conditions. When a fault occurs on the inner or outer race of a bearing, a series of impulses is generated in the vibration signal as the bearing defect interacts with another surface, and the impacts excite high-frequency resonances where the signal-to-noise ratio (SNR) is higher than the other frequency regions in the bearing system, thereby inducing a modulating phenomenon [1,2]. However, many other sources of bearing vibration such as the waviness of rolling elements etc. always result in the emergence of side bands around the principal bearing frequencies, which are more pronounced at higher frequencies [3]. Therefore, accurately

determining the high-frequency resonance band where the impulse occurs is key to successfully detecting bearing faults.

In earlier research, the resonance band is often determined by experimental tests, which are time-consuming [4]. As a statistical index, kurtosis is sensitive to the peaks caused by abnormal vibrations, and it is usually used as a direct measure of the transient impulses of the signal. Nevertheless, kurtosis is easily affected by noise. To overcome this limitation, Frequency Domain Kurtosis (FDK) was proposed by Dwyer [5] to complement the flaw of classical power spectral density (PSD), i.e., not being sensitive to the statistical nature of the signal. Inspired by this proposal, Spectral Kurtosis (SK) was presented by Antoni based on the Wold-Cramer theorem for non-stationary feature extraction in [6]. The basic idea of this approach is that the kurtosis at each frequency line of a signal is calculated to discover the presence of transients, and to indicate in which frequency bands these occur. For the convenience of industrial applications, Antoni further proposed the Fast Kurtogram (FK) using short-time Fourier transform (STFT) combined with 1/3 binary tree algorithms to split frequency bands to reduce computing time, as described in [7]. The fault impulses are extracted after a raw signal is processed by a band-filter for which the center frequency and bandwidth are optimized by FK. Since then, many studies have been conducted to enhance these theories [8–11]. Considering the problem whereby the parameters of the band-filter cannot be determined adaptively, Zhang et al. [12] combined genetic algorithms and FK to optimize the parameters. To extract transient impulsive signals under a low SNR condition, Wang et al. [13] proposed a time-frequency analysis method which combines the merits of ensemble local mean decomposition and FK to detect bearing faults. In [14], Lei proposed replacing STFT with wavelet package transform (WPT) to improve the kurtogram (WPTK). Recently, Wang proposed an enhanced kurtogram, in which the kurtosis values are calculated based on the envelope power spectrum of WPT nodes at different depths [15]. It is worth mentioning that each kurtosis value of the filtered signal is calculated without source identification in these methods, which is sometimes incorrect, especially when the vibration signal contains random knocks which usually have higher amplitudes, as well as kurtosis values which are far larger than those of real faults [4,16]. This effect means that the optimal frequency band corresponding to the maximum kurtosis is the resonance band containing random knocks, while the real fault signature is missing. To solve this problem, Barszcz et al. [17] proposed a higher resolution kurtosis index, the Protrugram, which is obtained by calculating the kurtosis of the envelope spectrum amplitudes of a narrow band filtered signal along the frequency axis. However, the optimal filter bandwidth depends on a certain knowledge of the sought fault. In [18], McDonald took advantage of the periodicity of the faults and proposed Correlated Kurtosis (CK) to detect cyclic transients. To make the extracted fault characteristic clear, they proposed an iterative selection process for the first and M-shift to maximize the CK. Combining CK and Redundant Second-Generation Wavelet Package Transform (RSGWPT), Chen proposed an improved kurtogram in [19]. In addition to these high-frequency resonance techniques, a non-resonance-based approach is desirable in an industrial environment, such as the Auto-regression moving average [20] and higher-order energy operator fusion methods [21]. The effectiveness of these methods has been verified when a shaft rotates at constant speeds. However, bearings usually operate at variable speed conditions in practice, which leads the fault features to no longer be discrete frequency lines, but rather, frequency bands related to the shaft rotating frequency [22–24]. During speed up and speed down processes, impulses induced by the faults are non-periodic in the time domain, which means that the method based on the indexes derived from the kurtosis index will be weakened in non-stationary feature extraction. Therefore, the question of how to recover the fault impulses from the signal collected in the varying speed conditions must be solved.

In this paper, a new index, Order Spectrum Correlation Kurtosis (OSCK) is proposed. By replacing the OSCK with the kurtosis in the FK, an improved kurtogram, Fast Order Spectrum Correlation Kurtogram (FOSCK) is constructed. In this method, the original non-stationary vibration signal is filtered by a 1/3 binary tree strategy, the envelope of each filtered signal is calculated by using the

Hilbert transform, and the trend term of each envelope is eliminated by a low-pass filter whose cutoff frequency is lower than the minimum of the rotating frequency. Then each envelope signal is resampled into a stationary one in the angular domain using the Computed Order Tracking (COT) technique to remove the effects of speed fluctuation. The OSCK of each resampling envelope signal is calculated and utilized to generate a diagram in which the frequency band corresponding to the highest value can then be considered for further analysis. A band-pass filter is set to maintain the desired band, and is used to extract the optimal frequency band signal. After that, the envelope of the filtered signal is resampled into angular domain by using the COT. The order structure of the angular domain envelope signal can be used to determine the type of fault by identifying its characteristic order. Compared with the FK, the WPTK and the Protrugram, the proposed method can extract bearing fault characteristic information more exactly under relatively large speed fluctuations and heavy interference environments.

## 2. Theoretical Background

### 2.1. Overview of Spectral Kurtosis and Fast Kurtogram

To overcome the shortcomings of the power spectral density (PSD), which is not sensitive to the statistical nature of the signals, the frequency domain kurtosis (FDK) was first introduced by Dwyer. It can highlight the frequency harmonic that is smeared because of random variation in the periodicity [5]. Inspired by this development, Antoni proposed spectral kurtosis (SK) in [6]. Different from FDK, which calculates the kurtosis of a particular frequency's amplitude, SK calculates the kurtosis of the complex envelope of filtered signals [17].

According to the Wold-Cramér decomposition theorem, a zero-mean non-stationary signal $x(n)$ can be expressed as [7,25]:

$$x(n) = \int_{-l/2}^{+l/2} H(n,f)e^{j2\pi fn}dX(f) \tag{1}$$

where $dX(f)$ is a spectral increment and $H(n,f)$ is the complex envelope of $x(n)$ at frequency $f$. The SK can be defined as the fourth-order normalized cumulant [7]:

$$K_x(f) = \frac{\left\langle |H(n,f)|^4 \right\rangle}{\left\langle |H(n,f)|^2 \right\rangle^2} - 2 \tag{2}$$

where the symbol $\langle\ \rangle$ denotes the temporal average operator. The constant $-2$ is used here because $H(n,f)$ is complex. Considering the presence of added noise, the SK of the non-stationary process $x(n)$ is described by:

$$K_y(f) = \frac{K_x(f)}{[1+\rho(f)]^2} \tag{3}$$

where $\rho(f)$ is the noise-to-signal ratio at frequency $f$. The transients in signals increase the spectral kurtosis value. Therefore, SK possesses the ability to detect and localize the presence of transients from a signal. However, to detect a narrow-band transient buried in noise, SK depends both on frequency and frequency resolution. Although this can be performed by computing the SK of each combination of different central frequencies $f_c$ and bandwidths $(\Delta f)_m$, this process is time consuming. To improve the calculation efficiency, Antoni utilized 1/3 binary tree algorithms to split the frequency band; then, the SK of each level $m$ and bandwidth $(\Delta f)_m$ were calculated to construct a 3D map-fast kurtogram, as shown in Figure 1. The horizontal axis represents the frequency, the vertical axis represents the level of the split frequency band, and the third dimension represents the color band, which is the kurtosis value of the filtered signal's envelope spectrum for each frequency bandwidth. The node with the largest kurtosis is chosen as the optimal band. However, the fast kurtogram calculates each kurtosis value of the filtered signal without source identification, which is sometimes incorrect.

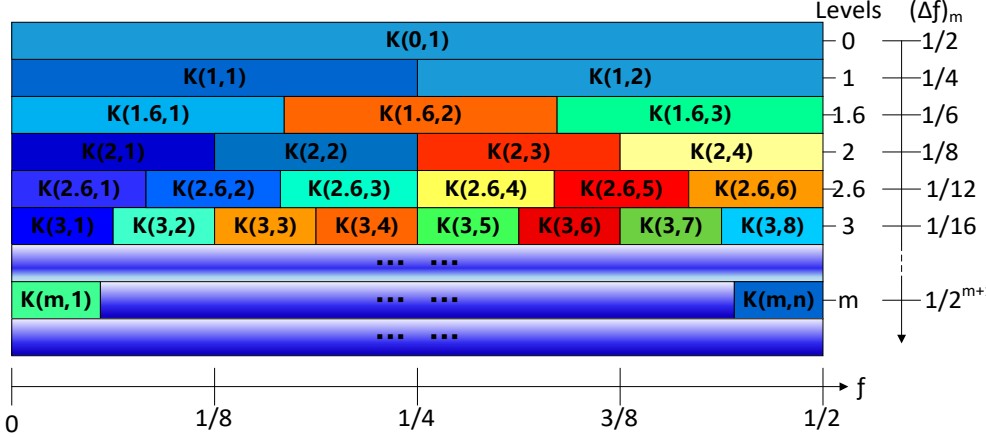

**Figure 1.** The paving of the Fast Kurtogram.

### 2.2. Order Spectrum Correlated Kurtosis

In [18], McDonald found that the kurtosis value of a signal with a single impulse is always higher than a signal containing consecutive periodicity of impulses. To solve the effect of random transients on kurtosis, McDonald proposed Correlated Kurtosis (CK) to detect signal cyclic transients by using the periodicity of the fault nature, and verified that CK can decrease the inference of single transient impulses. The CK of a vibration signal $x$ is defined as [14]:

$$CK(x) = \frac{\sum_{n=1}^{N} \left( \prod_{m=0}^{M} x_{n-mT} \right)^2}{\left( \sum_{n=1}^{N} x_n^2 \right)^{M+1}} \tag{4}$$

where $N$ is the length of $x$, $T$ is the period of interest impulses, and $M$ is the CK shift.

In [26], it was also observed that the kurtosis value is considerably affected by shaft rotational frequency. To eliminate the influence of speed on kurtosis and the frequency smear, COT is employed to convert the non-stationary filtered envelope time signal into the stationary vibration in the angular domain. Based on the key-phase signal, which is used to obtain the sampling-time marks of the even-angle sampling, an interpolation scheme is employed for resampling the original time-domain signal into the angular domain. Here, we use cubic spline interpolation. After that, the envelope order spectrum is utilized to expose the order structure in the signal, and the fault characteristic order (FCO) can be indicated clearly.

Previous research found that kurtosis is sensitive to external interference, especially in varying speed conditions, and the components contained in the signal have a single transient characteristic. Thus, it is difficult to detect fault sensitive components. To address the above problem, the order spectrum analysis is combined with correlated kurtosis to form a new feature, OSCK, to detect the fault-sensitive frequency band under varying speed conditions.

The OSCK can be defined as follows:

$$OSCK(A, T) = \frac{\sum_{n=1}^{N} \left( \prod_{m=0}^{M} A_{n-mT_o} \right)^2}{\left( \sum_{n=1}^{N} A_n^2 \right)^{M+1}} \tag{5}$$

where $A$ is the envelope order spectrum amplitudes, and $T_o$ is the period of impulses.

### 2.3. Procedure of the Proposed Method

Based on the discussion above, an improved kurtogram is proposed for rolling bearing fault diagnosis under varying speed conditions. The optimal frequency band corresponding to the maximum OSCK in the kurtogram is filtered; then, the fault can be identified by envelope order spectrum analysis of the filtered signal. The scheme of the proposed method is shown in Figure 2, and the details are described as follows:

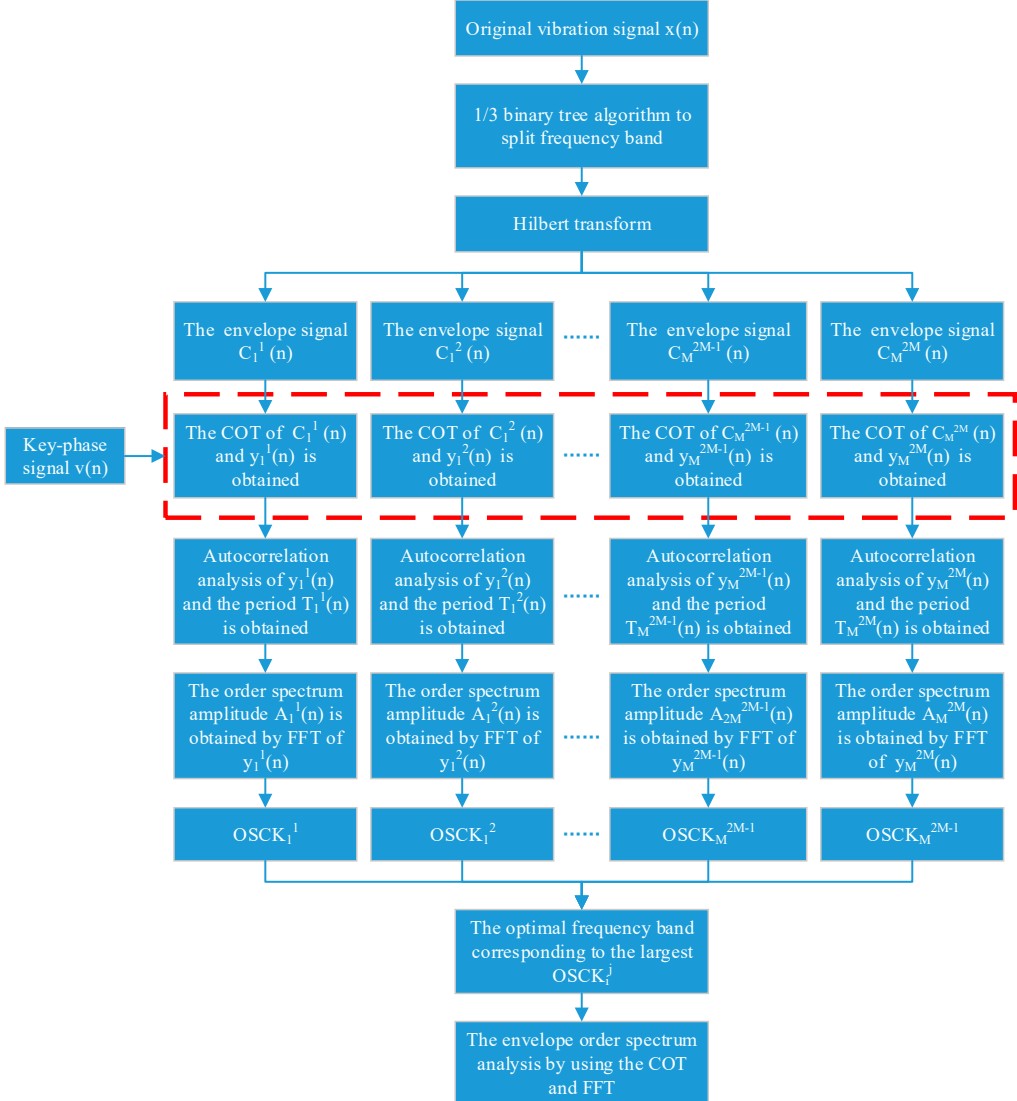

**Figure 2.** The flowchart of proposed method.

Step 1. The original vibration signal $x(n)$ and synchronous sampling key-phase signal $v(n)$ measured by different accelerometers are loaded.

Step 2. The signal $x(n)$ is filtered with a 1/3 binary tree strategy. Let $h(n)$ be a low-pass prototype filter, and two quasi-analytic low-pass and high-pass analysis filters $h_l(n)$ and $h_h(n)$ are constructed, which have the frequency bands [0; 1/4] and [1/4; 1/2], respectively:

$$h_l(n) = h(n)e^{j\pi n/4} \tag{6}$$

$$h_h(n) = h(n)e^{j3\pi n/4}, j^2 = -1 \tag{7}$$

Different central frequency $f_{ci}$ and bandwidth $(\Delta f)_m$ corresponding signals are iteratively obtained by using these filters in a pyramidal manner, which has tree-structured filter-banks and denote as $x_m^i(n)$, where $i = 1, 2, \cdots, 2^m$, $m = 0, 1, \cdots, M-1$, and $M$ is the largest decomposition level [7]. The envelope of each filtered signal $C_m^i(n) = |x_m^i(n) + jHil[x_m^i(n)]|$ can be created, where $Hil$ is the Hilbert transform and the symbol $||$ represent the absolute value. The trend term of each envelope is eliminated by a low-pass filter whose cutoff frequency is lower than the minimum of the rotating frequency.

Step 3. Each filtered envelope signal $C_m^i(n)$ is resampled in the angular domain. Each filtered envelope signal is non-stationary in the time domain due to the variable speed operations that cause spectrum smearing and low autocorrelation. To solve this problem, COT is employed to convert the non-stationary envelope signal into the stationary envelope signal in the angular domain by using the key-phase signal $v(n)$, whose length is the same as the original signal $x(n)$. The resampling envelope signal denote as $y_m^i(n)$.

Step 4. The OSCK of each resampling envelope signal is calculated. First, the autocorrelation analysis of $y_m^i(n)$ is performed to enhance the involved periodic impulsive feature and the autocorrelation coefficient can be calculated by the following formula:

$$R_t = \frac{\left| \sum_{j=1}^{n} \left[ y_{m,j}^i - \bar{y}_m^i \right] \times \left[ y_{m,j+t}^i - \bar{y}_m^i \right] \right|}{\sqrt{\sum_{j=1}^{n} \left[ y_{m,j}^i - \bar{y}_m^i \right]^2 \times \sum_{j=1}^{n} \left[ y_{m,j+t}^i - \bar{y}_m^i \right]^2}} \tag{8}$$

where $R_t$ is the autocorrelation coefficient, $\bar{y}_m^i$ is the average value of signal $y_m^i(n)$, and $t$ denotes the length of the delay. Through autocorrelation operation, the periodic impulsive signal component related to the bearing fault is strengthened. The period $T_o$ of impulses of interest is denoted as:

$$T_o = \text{argmax}(R_t) \tag{9}$$

Second, the order spectrum of $y_m^i(n)$ is obtained by Fourier transform and the order spectrum amplitude denote as $A_m^i(n)$. Last, the OSCK values are calculated using Equation (5). The OSCK values of all nodes are represented in the kurtogram.

Step 5. The frequency band corresponding to the maximum OSCK value are filtered by a band-pass filter, and the envelope of the filtered signal is transformed into angular domain by using the COT, and the envelope order spectrum is used to map the angle domain signal to the order-dependent signal to identify the bearing fault characteristic order, which is usually calculated by Equations (10)–(12). The outer race fault characteristic order $FCO_o$, the inner race fault characteristic order $FCO_i$ and the rolling element fault characteristic order $FCO_b$ are formulated as follows:

$$FCO_o = \frac{Z}{2} \left( 1 - \frac{d}{D} \cos \alpha \right) \tag{10}$$

$$FCO_i = \frac{Z}{2} \left( 1 + \frac{d}{D} \cos \alpha \right) \tag{11}$$

$$FCO_b = \frac{Z}{2d} \left( 1 - (\frac{d}{D})^2 \cos^2 \alpha \right) \tag{12}$$

where $Z$ is the number of rolling elements, $\alpha$ is the contact angle, and $d$ and $D$ are the diameter of the rolling element and pitch diameter, respectively.

## 3. Simulations

In this section, several simulations are used to demonstrate the effectiveness of the proposed method. Considering the complexity of the rotating system, the synthetic signals usually include three terms: deterministic components, including the fundamental frequency and harmonics of the shaft, which are caused by factors such as misalignment, eccentricity or imbalance. Random components, which represent a series of impulses excited by a fault, and measurement noise. The simulated signal is defined as:

$$x(t) = \underbrace{\sum_m A_m \cos(2\pi m f(t)t + \phi_m)}_{\text{the deterministic components}} + \underbrace{[1 + \lambda M(t)] * \sum_n B_n s(t - t_n - \tau_n)}_{\text{the random components}} + \underbrace{n(t)}_{\substack{\text{the noise} \\ \text{components}}} \tag{13}$$

where $A_m$ and $\phi_m$ are the amplitude and initial phase of the $m$th harmonic frequency of the shaft, respectively; $f(t)$ is the instantaneous rotating frequency of the shaft; $1 + \lambda M(t)$ denotes the amplitude modulation term,

$$1 + \lambda M(t) = \begin{cases} 1, \lambda = 0 & \textit{if bearing outer race fault} \\ 1 + \lambda \cos(2\pi f t), \lambda \neq 0 & \textit{if bearing inner race fault} \\ 1 + \lambda \cos(2\pi f_{cage} t), \lambda \neq 0 & \textit{if bearing rolling element fault} \\ 0 & \textit{normal} \end{cases} \tag{14}$$

where $f_{cage}$ is the cage speed; $B_n$ and $t_n$ are the amplitude and occurrence time of the nth impulse, and the occurrence time $t_n$ is determined according to the instantaneous rotating frequency $f(t)$ and the fault order frequency $f_o$; $\tau_n$ is the coefficient used to calculate slippage time, which varies from 1% to 2% of the time period of the fault impulse; $s(t)$ is the impulse response function of the system; and $n(t)$ is the Gaussian white noise that is uncorrelated with other components. The impulse response function can be written as

$$s(t) = \begin{cases} e^{-\beta(t - t_n - \tau_n)} \sum_i \sin\{2\pi f_r(t - t_n - \tau_n)\}, & \textit{if } t - t_n > 0 \\ 0 & \textit{otherwise} \end{cases} \tag{15}$$

where $\beta$ is the structural damping coefficient, and $f_r$ is the resonance frequency.

### 3.1. Simple Simulation for the Study of COT Analysis after Time-Domain Filtering

In the FOSCK, the COT must be used after the envelope demodulation analysis. In order to explain this and determine the influence of COT on impulse feature extraction, a simple outer race simulated signal that has a single resonant frequency and consists of a series of pure impulses is shown in Figure 3. The simulation signal parameters in the model are given in Table 1, where $f_s$ is sampling frequency

**Table 1.** Parameters of the simulation model.

| N (s) | $B_n$ | $\varphi_m$ | f (Hz) | $f_o$ | $f_s$ (kHz) | $f_r$ (kHz) | β (kHz) | $\tau_n$ |
|---|---|---|---|---|---|---|---|---|
| 3 | 1 | 0 | 5–15 | 4.5 | 20 | 5 | 1.2 | 0.01 |

The rotating frequency is given by

$$f(t) = 10 + 5 \times \sin(10\pi t) \tag{16}$$

and illustrated in Figure 3a. The corresponding time-domain signal and angle-domain resampling signal are shown in Figure 3b,c, respectively. It is clear that the intervals of adjacent impulse responses

change synchronously with the rotating frequency in the time-domain signal; the rotating frequency is larger, and the adjacent impulse intervals are smaller. Unlike that of the time-domain signal, the pulse interval is unchanged in the angle-domain signal.

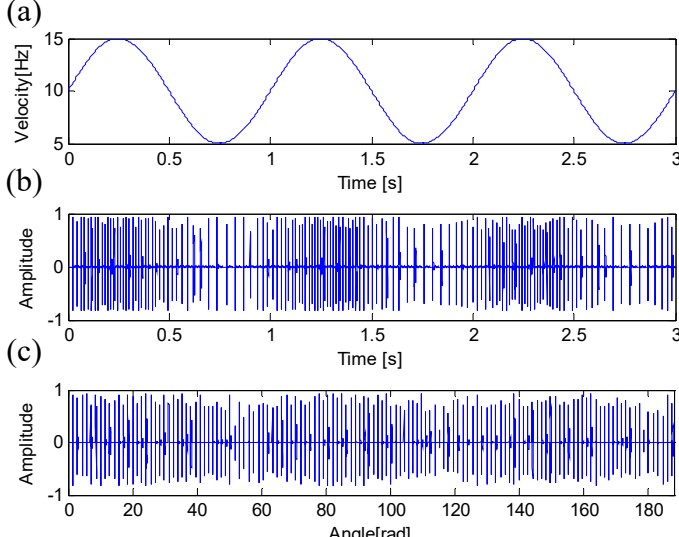

**Figure 3.** Simple simulation: (**a**) the shaft rotational frequency, (**b**) the impulse signal in the time domain and (**c**) the resampling signal in the angle domain.

The short-time Fourier transforms (STFTs) of the time domain and angle domain impulse responses are shown in Figure 4a,b, respectively. The carrier frequencies of the time-domain signal are concentrated around the resonant frequency, while the carrier orders of the angle-domain signal spread to a wider order scope. Therefore, it can be concluded that the COT procedure causes distortion of the signal resonance band, which is very important in the resonance demodulation analysis.

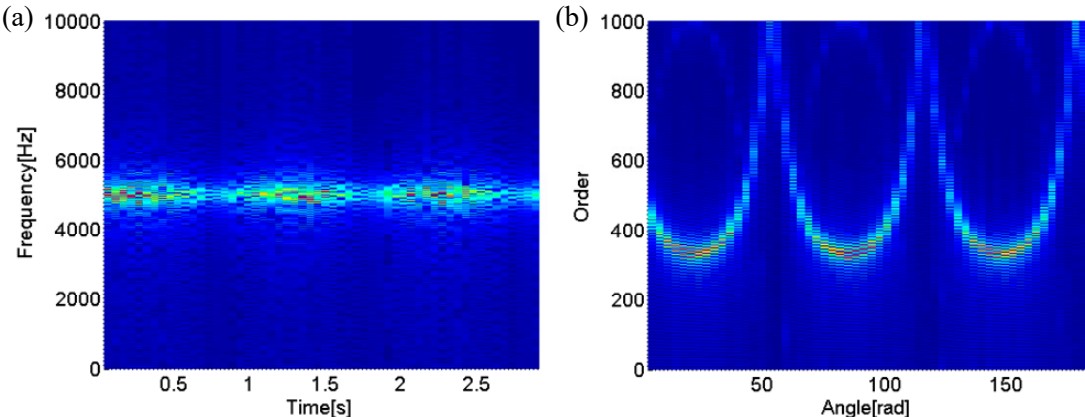

**Figure 4.** STFTs of the simulation: (**a**) STFT of the time-domain signal and (**b**) STFT of the angular-domain signal.

In addition to the above defects, as stated by A.B. Ming in [27], a simple filter with a fixed cutoff order cannot deal with the angle-domain signal, whose carrier orders vary over time. The envelopes obtained by a low pass filter with a fixed cutoff frequency in the time domain and a fixed cutoff order in the angle domain, as shown in Figure 5a,b, respectively. The amplitude of the filtered signal in the angle domain is distorted. Therefore, an envelope demodulation analysis must first be carried out in the time domain, then the COT method is performed on the envelope.

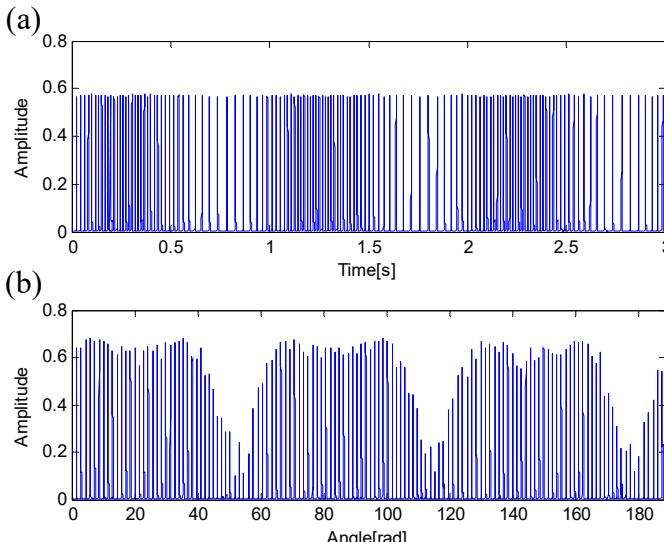

**Figure 5.** Envelopes of the filtered signals: (**a**) envelope of the time-domain filtered signal and (**b**) envelope of the angular-domain filtered signal.

### 3.2. Simple Simulation of the Influence of Rotational Speed Indication

In this simulation, to illustrate the effect of shaft rotational frequency on the different index values intuitively, the rotating frequency is given by

$$\begin{cases} f(t) = 10, & t = 0 \sim 3s \\ f(t) = 10 + 5 \times (t - 3), & t = 3 \sim 6s \\ f(t) = 25, & t = 6 \sim 9s \end{cases} \tag{17}$$

The other parameters are the same as those mentioned above. The simulation signal that only contains the pure impulsive signal is shown in Figure 6.

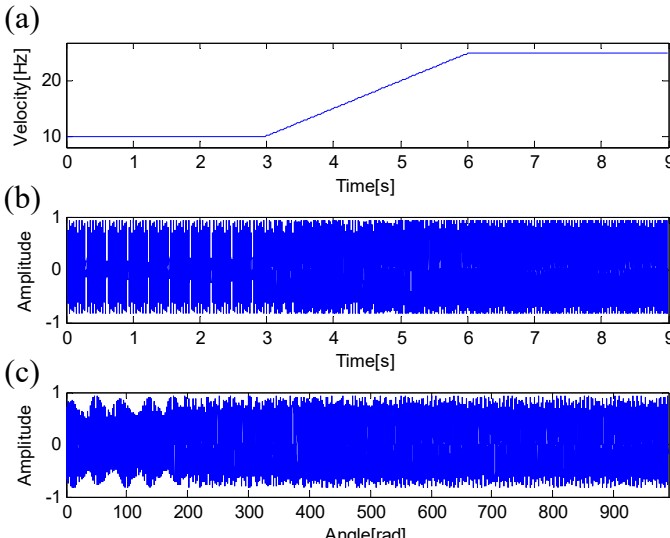

**Figure 6.** Simple simulation: (**a**) the shaft rotational frequency, (**b**) the impulse signal in the time domain and (**c**) the resampling signal in angle domain.

It is generally assumed that a high kurtosis value is treated as a sign of the presence of faults in a rotating mechanical system. However, this assumption has no application to the varying speed case. To demonstrate this special condition, both of the time domain signal and angular domain signal are

equally divided into 45 signal segments. A different index value of every signal segment is calculated respectively, and normalized to construct a vector, as shown in Figure 7, where the blue and red lines are denote the normalized kurtosis value and the normalized CK value of each time domain signal segment. The green and black lines are CK and OSCK values correspond to the angle-domain signal segments. Overall, the CK and kurtosis of the signal segment decrease as the speed increases, both in the time domain and the angle domain. It is worth noting that the OSCK of the angle domain signal segments is sensitive to the speed fluctuation while not being affected by the size of the speed. Therefore, the OSCK index is more suitable for the extraction of resonance bands under varying speed conditions.

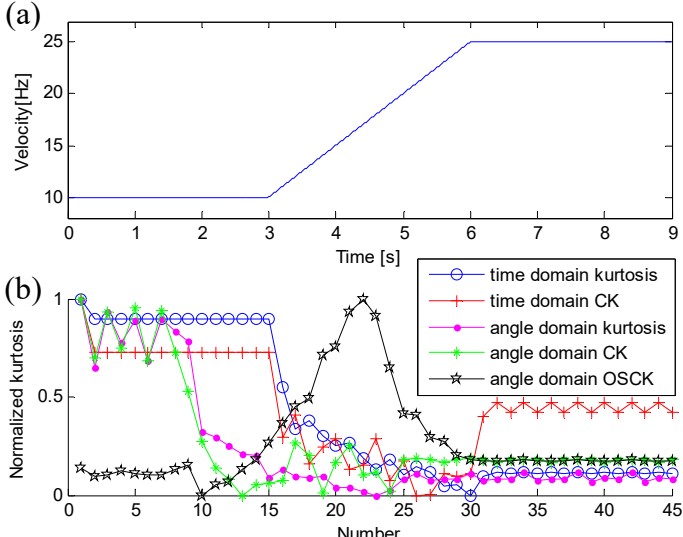

**Figure 7.** (**a**) The shaft rotational frequency, (**b**) the normalized kurtosis of different indexes.

### 3.3. Simple Simulation of the Influence of Random Shocks Indication

A high kurtosis value is often treated as a sign of the presence of faults in bearing fault diagnosis. However, the kurtosis value of a signal with a single impulse is always higher than a signal containing consecutive periodicity of impulses. To illustrate the influence of random shocks on a chosen resonance band, two cases are considered here. The parameters of the simulated signal are shown in Table 2.

**Table 2.** Parameters of the simulation model.

|  | N (s) | $B_n$ | $\varphi_m$ | $f$ (Hz) | $f_o$ | $f_s$ (kHz) | $f_{r1}$ (kHz) | $f_{r2}$ (kHz) | $\beta_1$ (kHz) | $\beta_2$ (kHz) | $\tau_n$ | SNR (dB) |
|---|---|---|---|---|---|---|---|---|---|---|---|---|
| Case 1 | 1 | 1 | 0 | 10–12 | 4.5 | 20 | 5 | \ | 1.2 | 3 | 0.01 | −5 |
| Case 2 | 1 | 1 | 0 | 10–12 | 4.5 | 20 | 5 | 7.3 | 1.2 | 3 | 0.01 | −5 |

3.3.1. Case 1: The Random Shocks Have the Same Resonant Frequency as the Fault Impulses

The noise-free simulated mixed signal, which contains fault impulses and a random shock, its noise-added signal and their frequency spectrums are shown in Figure 8.

The FOSCK of the simulated signal is paved in Figure 9a; the maximum OSCK is calculated at the 5.5th decomposition level and its corresponding frequency band is (4792, 5000) Hz. The corresponding envelope of the filtered signal and its resampling envelope signal are shown in Figure 9b,c, respectively. The envelope order spectrum is shown in Figure 9d. It can be observed that the fault characteristic order and its harmonics are quite efficiently extracted.

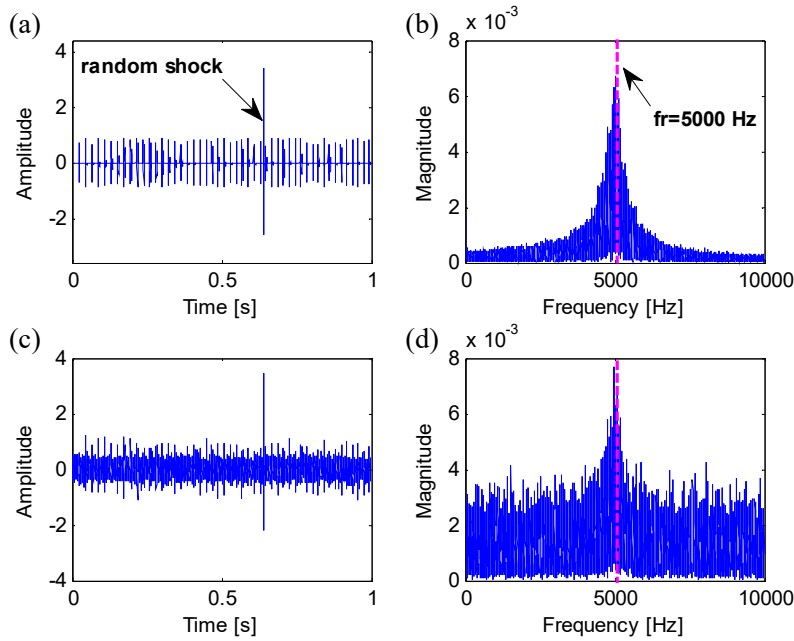

**Figure 8.** Simple simulation: (**a**) the simulated signal, (**b**) the frequency spectra of (**a**), (**c**) the noise-added signal with SNR = −5 dB and (**d**) the frequency spectra of (**c**).

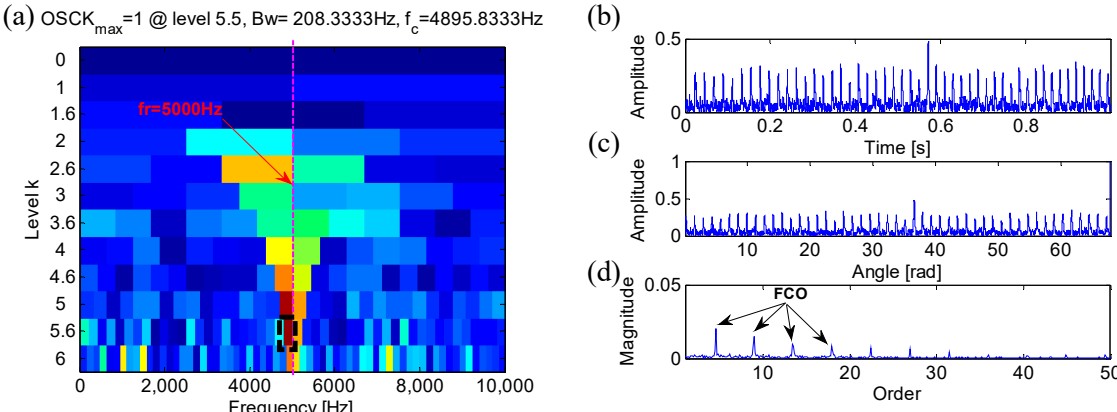

**Figure 9.** The results obtained by the FOSCK for processing the mixed signal with same resonant frequency: (**a**) FOSCK, (**b**) the envelope of the band-pass filtered signal, (**c**) the resampling envelope signal of (**b**,**d**) the envelope order spectrum of (**c**).

### 3.3.2. Case 2: The Random Shocks Have Different Resonant Frequencies from the Fault Impulses

In this simulation, a random shock with a different resonance frequency from the fault impulses is added to the pure signal; its noise-added signal is shown in Figure 10.

The paving of the FOSCK is shown in Figure 11a. The same to case 1, the optimal frequency band corresponding to the maximum CK is calculated at the 5.5th decomposition level and its frequency band is (4792, 5000) Hz. The envelope of the filtered signal and its resampling envelope signal are shown in Figure 11b,c, respectively. The envelope order spectrum is shown in Figure 11d, in which it the fault characteristic order is obvious.

Therefore, when dealing with a vibration signal with random shock interference, whether the random shock has the same resonance frequency band as the fault impulses or not, the proposed index OSCK can locate the fault resonant frequency band exactly.

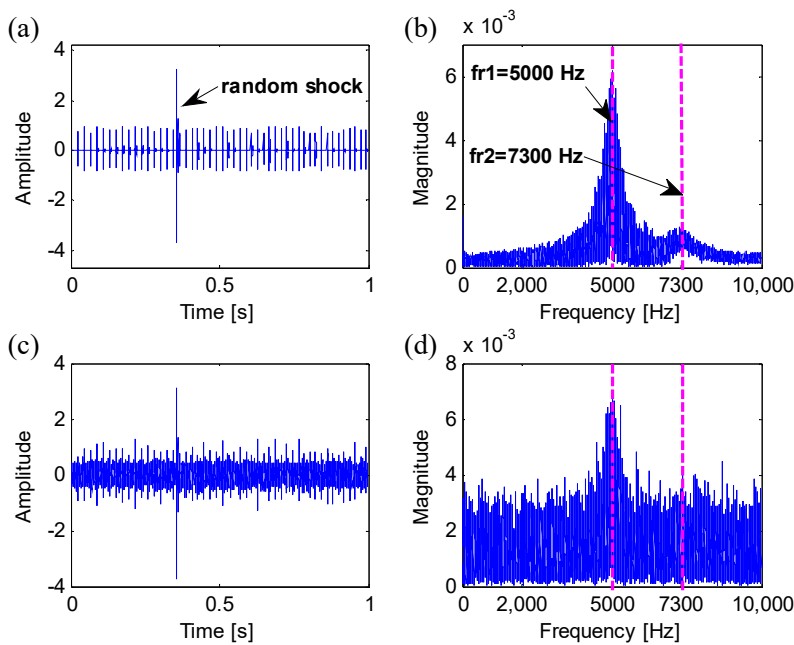

**Figure 10.** Simple simulation: (**a**) the simulated signal, (**b**) the frequency spectra of (**a**), (**c**) the noise-added signal with SNR= −5 dB and (**d**) the frequency spectra of (**c**).

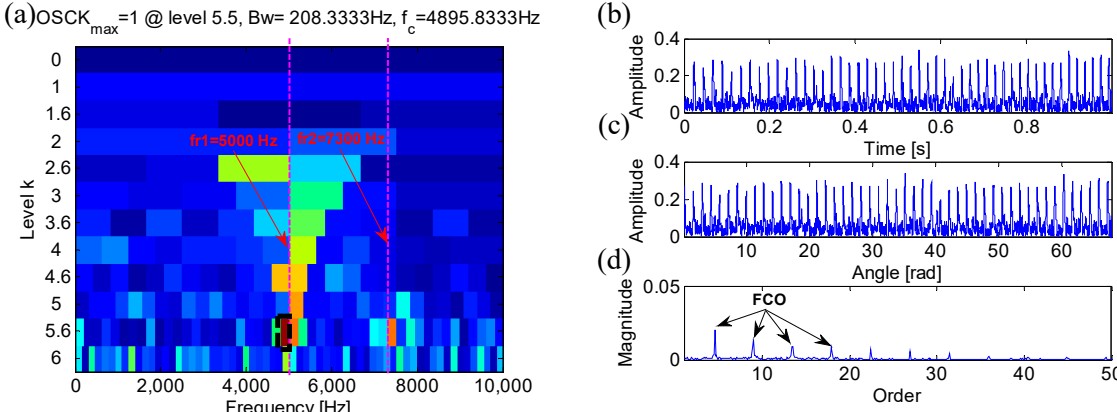

**Figure 11.** The results obtained by the FOSCK for processing the mixed signal with different resonant frequencies: (**a**) FOSCK, (**b**) the envelope of the band-pass filtered signal, (**c**) the resampling envelope signal of (**b,d**) the envelope order spectrum of (**c**).

### 3.4. Simple Simulation of the Influence of Multiple Impact Sources

To match the simulation closer to the real situation, a deterministic component and two random shocks are added to a fault impulse signal, and a considerable amount of Gaussian noise is added too. The simulation signal parameters are given in Table 3. The different components and their frequency spectra are shown in Figure 12.

**Table 3.** Parameters of the simulation model.

| N (s) | $B_n$ | $\varphi_m$ | $f$ (Hz) | $f_o$ | $f_s$ (kHz) | $f_{r1}$ (kHz) | $f_{r2}$ (kHz) | $\beta_1$ (kHz) | $\beta_2$ (kHz) | $\tau_n$ | SNR (dB) |
|---|---|---|---|---|---|---|---|---|---|---|---|
| 1 | 1 | 0 | 10–15 | 4.5 | 20 | 5 | 7.3 | 1.2 | 3 | 0.01 | −5 |

The FK, WPTK and Protrugram are applied to analyze the mixed signal, and the results are shown in Figures 13–15, respectively. It is worth noting that in the following paragraphs, a wavelet packet basis

db10 is used in the WPTK as that given in [14], and a bandwidth (BW) that includes the 3rd harmonic of the characteristic frequency is selected in the Protrugram according to the rules mentioned in [28]. The analysis results show that both the FK and the WPTK failed to detect the fault-sensitive resonance frequency, and their optimal frequency bands were located near 7300 Hz, which corresponded to the random shock. From the envelopes of the filtered signal shown in Figures 13 and 14b,c, the amplitudes of the fault impulses are relatively small, while the random shocks are obviously. The FCO and its harmonics are difficult to identify from the envelope order spectra, as shown in Figures 13 and 14d. Differing from the FK and WPTK, the Protrugram is shown in Figure 15a with BW equals to 300 Hz and the step is 50 Hz. The optimal frequency band relevant to the maximum kurtosis is the fault-sensitive frequency band. The corresponding envelope of the filtered signal and its resampling envelope signal are shown in Figure 15b,c, respectively. The FCO and its harmonics are clearly shown in Figure 15d. The FOSCK is shown in Figure 16a, the optimal frequency band corresponding to the maximum OSCK is calculated at the 5.5th decomposition level, and its frequency band is (4792, 5000) Hz. The fault impulses are clear in the envelopes shown in Figure 16b,c. The FCO and its harmonics are clearly visible in Figure 16d, which means the FOSCK is robust to the varying speed and random shocks.

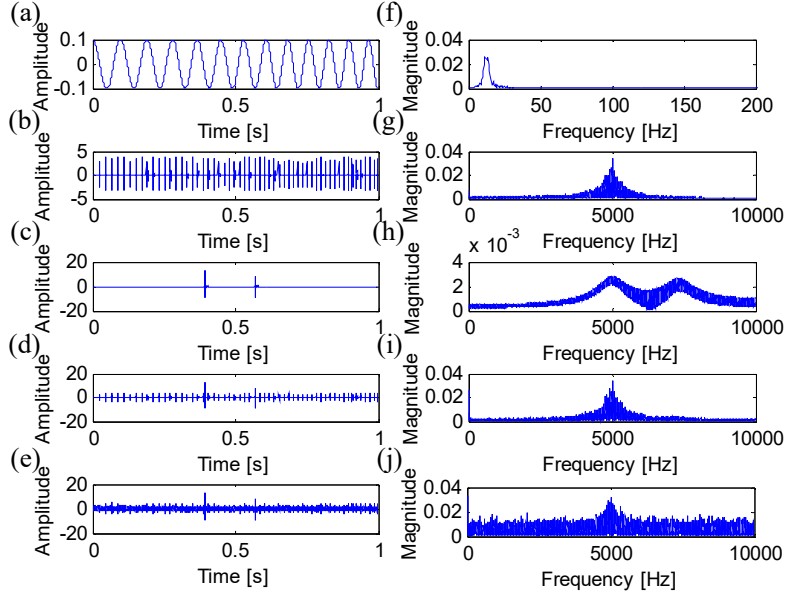

**Figure 12.** Simulated signals: (**a**) a deterministic component signal, (**b**) the fault impulse signal, (**c**) the random shocks, (**d**) the synthetic signal without noise added, (**e**) the synthetic signal with noise-added and SNR = −5 and (**f**–**j**) the frequency spectra of the different components.

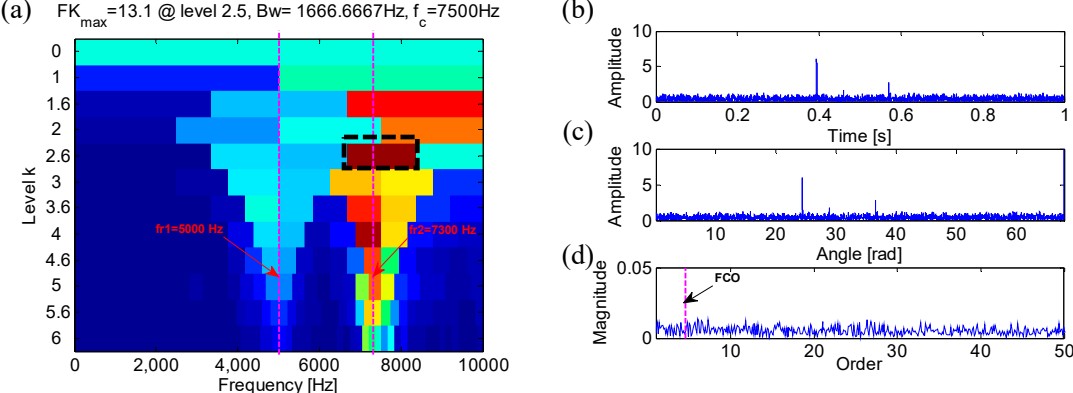

**Figure 13.** The results obtained by the FK for processing the mixed signal: (**a**) FK, (**b**) the envelope of the band-pass filtered signal, (**c**) the resampling envelope signal of (**b**,**d**) the envelope order spectrum of (**c**).

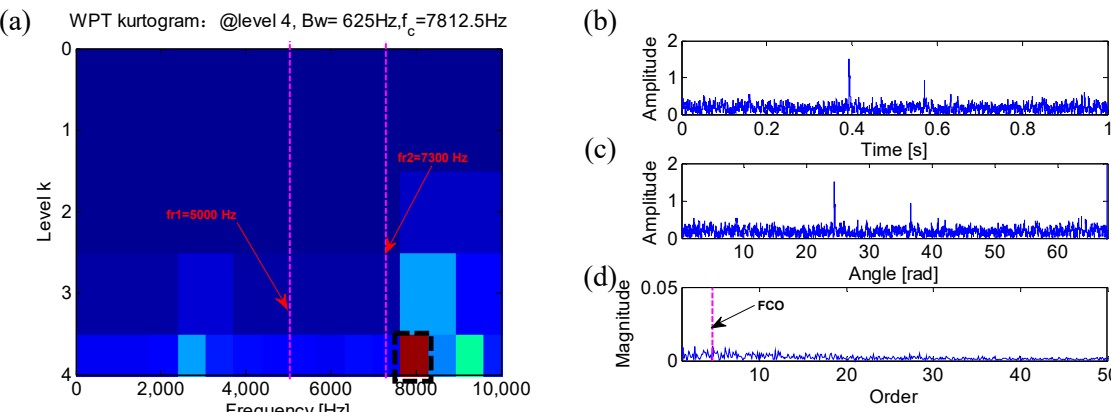

**Figure 14.** The results obtained by the WPTK for processing the mixed signal: (**a**) WPTK, (**b**) the envelope of the band-pass filtered signal, (**c**) the envelope of the resampling of (**b**,**d**) the envelope order spectrum of (**c**).

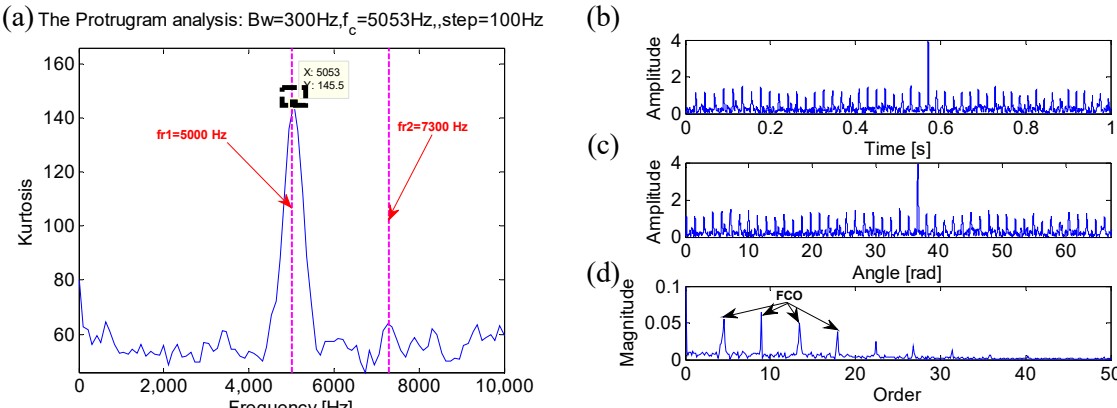

**Figure 15.** The results obtained by the Protrugram for processing the mixed signal: (**a**) Protrugram, (**b**) the envelope of the band-pass filtered signal, (**c**) the envelope of the resampling of (**b**,**d**) the envelope order spectrum of (**c**).

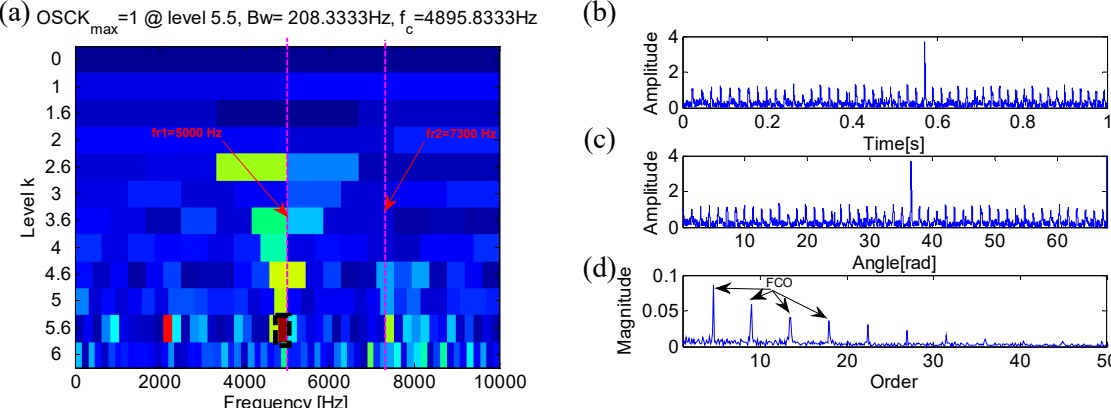

**Figure 16.** The results obtained by the FOSCK for processing the mixed signal: (**a**) FOSCK, (**b**) the envelope of the band-pass filtered signal, (**c**) the envelope of the resampling of (**b**,**d**) the envelope order spectrum of (**c**).

## 4. Experimental Evaluation

To further examine the effectiveness of the proposed method, an experiment is carried out on a Spectra Quest Machinery Fault Simulator. The bearing test rig consists of an AC motor, a flexible

coupling to connect the shaft to the motor, a tachometer mounted on the motor, a rotor disk mounted onto the shaft, the outboard bearing housing, and two rolling element bearings. One of the bearings without defects is located in the bearing housing closer to the motor, and the other one is located farther from the motor. The ICP acceleration sensors are fixed on the bearing housing to collect vibration signals at a sampling frequency of 20 kHz. A data acquisition instrument and a computer are used for the analysis. The test rig is shown in Figure 17.

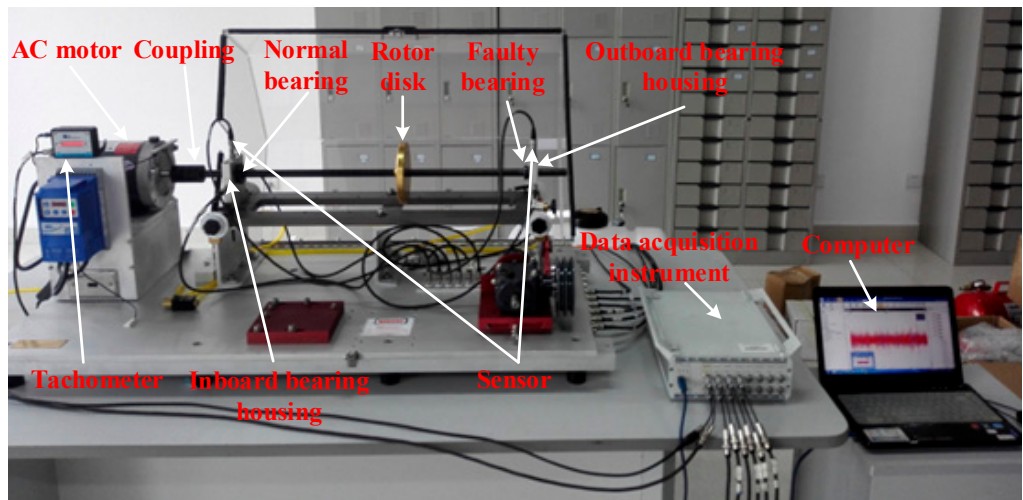

**Figure 17.** The test bench for bearing fault detection.

The parameters of the bearings are listed in Table 4.

**Table 4.** Parameters of the bearings.

| Fault Severity | Bearing Type | Number of Balls | Contact Angle | Pitch Diameter | Ball Diameter | BPFO | BPFI |
|---|---|---|---|---|---|---|---|
| 3/4" Rotor bearing | ER-12K | 8 | 0 | 1.318 in | 0.3125 in | 3.048 | 4.95 |

### 4.1. Normal Bearing

To illustrate the effectiveness of the proposed bearing fault diagnosis method, a baseline case is first studied, in which both bearings are healthy, as shown in Figure 17. Due to relative motion, bearing components generate vibrator signals in operation, as shown in Figure 18a. The shaft rotational speed is shown in Figure 18b. Figure 18c shows the frequency spectra of a healthy bearing as the shaft accelerates from 20 Hz to 25 Hz within 3.347 s, i.e., the acceleration a equals 3/2 Hz/s. The time-frequency representation (TFR) of the signal is obtained via STFT, as shown in Figure 18d. In the TFR, several suspected resonance frequency bands, in which the energy is most concentrated, are adaptively removed using different indexes in Figure 19. Based on the maximum of each index, an optimal frequency band is selected for further analysis. In Figure 19a, the optimal frequency band is (4375, 4687.5) Hz in the FK. The WPTK is shown in Figure 19b, and the corresponding optimal frequency band is (9376, 10,000) Hz at the 4th decomposition level. The Protrugram with BW = 400 Hz and step = 50 Hz is shown in Figure 19c, and the maximum kurtosis is calculated at 4974 Hz. In Figure 19d, the FOSCK, the optimal frequency band corresponding to the maximum CK is calculated at the 5th decomposition level, and its frequency band is (3125, 3437.5) Hz. The original signal is filtered by using different band-pass filters corresponding to these optimal frequency bands. The envelope of each filtered signal is calculated by using the Hilbert transform and resampled into the angular domain by using COT. The envelope order spectrum analysis result is shown in Figure 20. In all the figures, the dominant order components are related to the shaft rotational order (SRO) and its harmonics. These

results imply that both bearings are healthy. It is worth mentioning that the SRO and its harmonics appear due to rotor disk machining error and installation error.

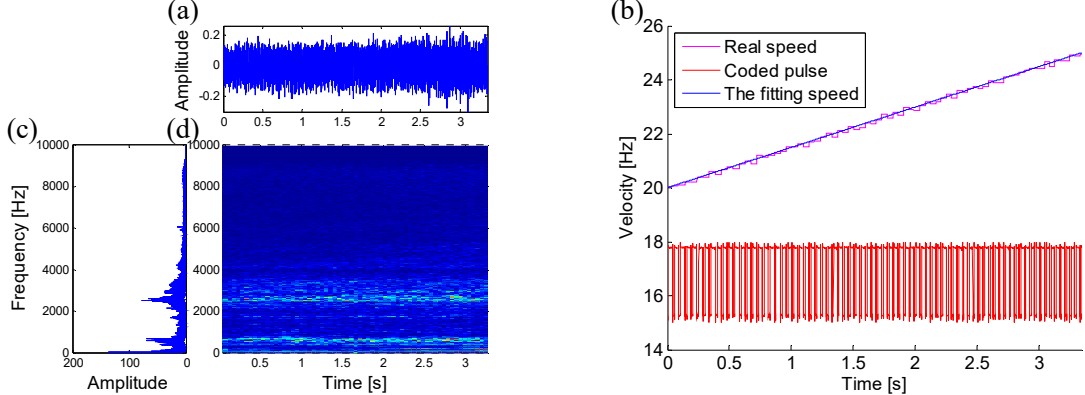

**Figure 18.** The signal measured from a normal bearing: (**a**) time-domain signal, (**b**) the shaft rotational frequency from 20 Hz to 25 Hz, (**c**) the frequency spectrum of (**a**,**d**) TFR by using STFT.

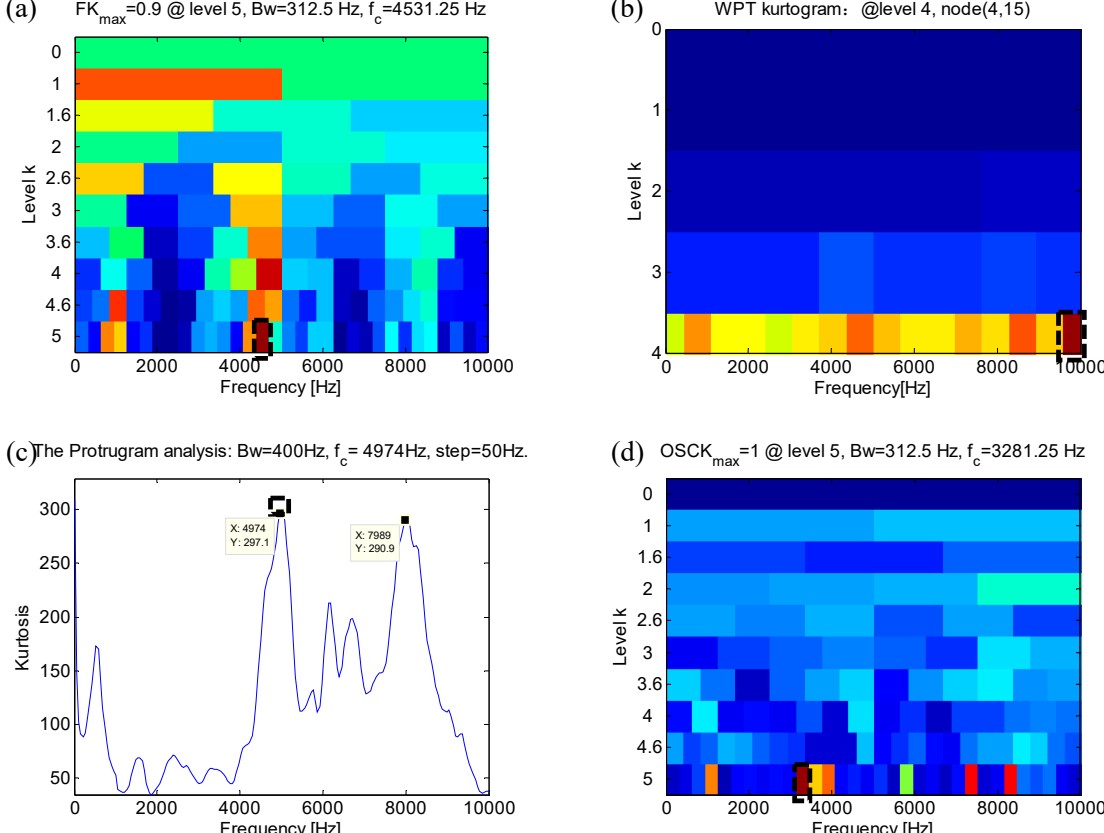

**Figure 19.** The optimal frequency band obtained by different methods: (**a**) FK, (**b**) WPTK, (**c**) Protrugram and (**d**) FOSCK.

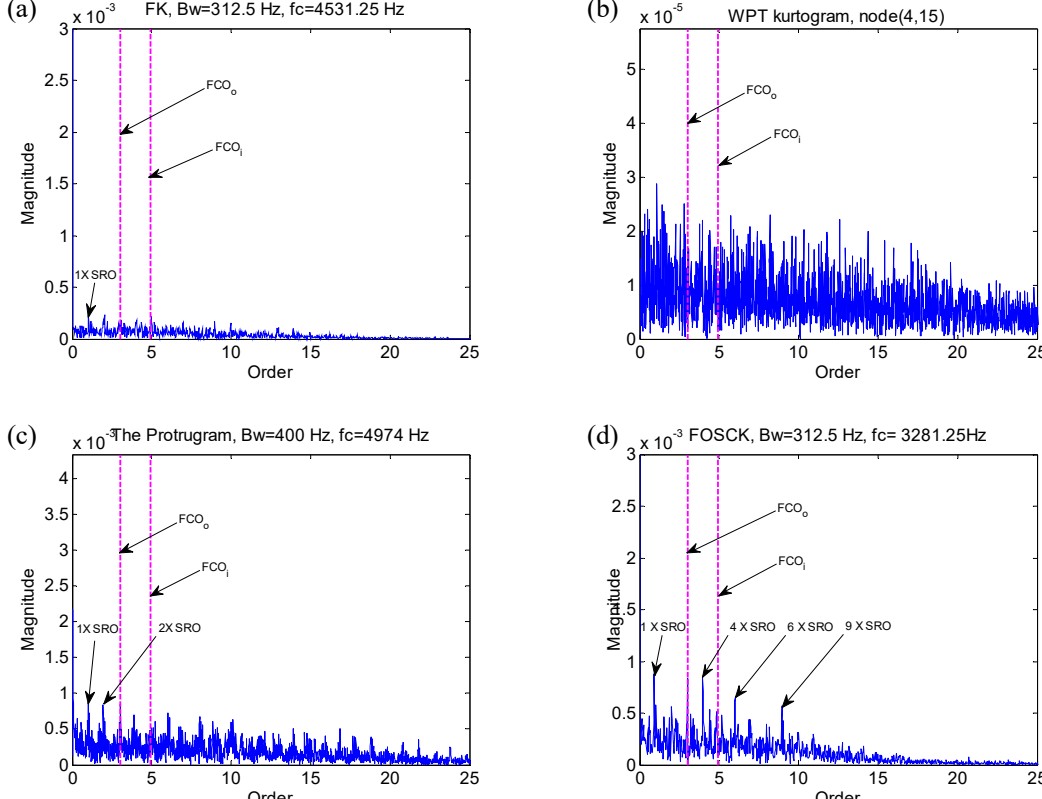

**Figure 20.** The envelope order spectra of the signal obtained by different methods: (**a**) FK, (**b**) WPTK, (**c**) Protrugram and (**d**) FOSCK.

### 4.2. Diagnosis of a Bearing with an Outer Race Fault

In the outer race fault case, the right bearing is replaced by an outer race defect, as shown in Figure 17. Aiming to demonstrate that the method is robust to rotational speed changes and random shock interference, the vibration signals under three different acceleration conditions are analyzed, as illustrated in Table 5. In addition, white Gaussian noise (SNR = −3 dB) is added to the collected signal to make the detection more challenging.

**Table 5.** Parameters of each experiment.

| Acceleration (Hz/s) | Experimental Study #1 | Experimental Study #2 | Experimental Study #3 |
|---|---|---|---|
| a | 4/3 | 3/2 | 3 |

#### 4.2.1. Experimental Study #1

The collected vibration signal and the rotating speed are shown in Figure 21a,b. The frequency spectrum of the vibration signal is shown in Figure 21c, in which spectrum smearing could be observed due to the variable rotating speeds. In addition, the TFR of the signal is blurry and lacks detail due to background noise interference, which may come from other coupled machine components and the working environment, making it more difficult to identify the fault type in Figure 21d.

Figure 22 shows the signal analysis results for the outer race fault case when a is equal to 4/3 Hz/s. The FK is paved in Figure 22a, in which the optimal frequency band is (2916.67, 3333.33) Hz. Figure 22b shows the WPTK, in which the maximum kurtosis is calculated at the 4th decomposition level, and its corresponding optimal frequency band is (3125, 3750) Hz. The Protrugram is shown in Figure 22c and the center frequency is 638.3 Hz. Figure 22d gives the FOSCK, in which the maximum OSCK is calculated at the 3.5th level, and the optimal frequency band is (3333.33, 4166.67) Hz. Different

band-pass filters are used to filter out the corresponding frequency band signals, and their envelopes are calculated using the Hilbert transform. Then, each envelope of these filtered signals is resampled into the angular domain by using COT, and the envelope order spectrum analysis results are shown in Figure 23a–d, respectively. It is clear that all the filtered signals contain fault components, which also verifies that the fault impulse has broadband characteristics. In Figure 23, the FCO of the bearing outer race fault and its triple octaves are very clear. Therefore, all the methods mentioned above can effectively detect the bearing outer race fault, as in the case of acceleration a = 4/3 Hz/s.

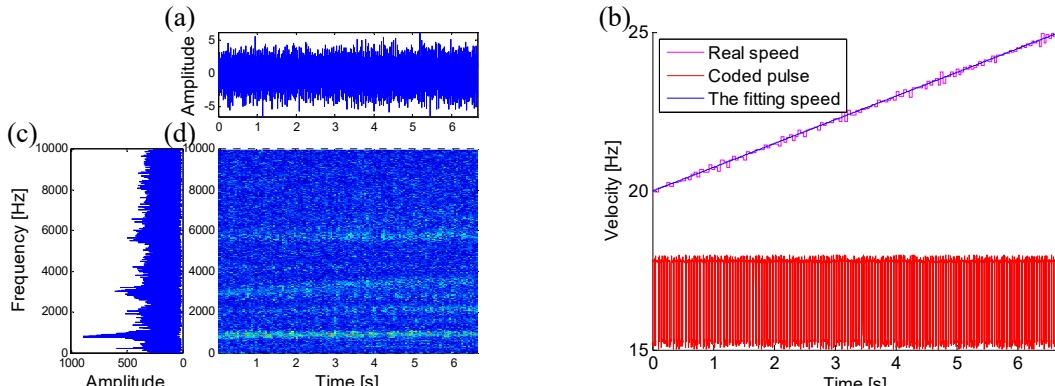

**Figure 21.** The signal measured from an outer race fault bearing: (**a**) time-domain signal, (**b**) the shaft rotational frequency from 20 Hz to 25 Hz, (**c**) the frequency spectrum of (**a**,**d**) TFR by using STFT.

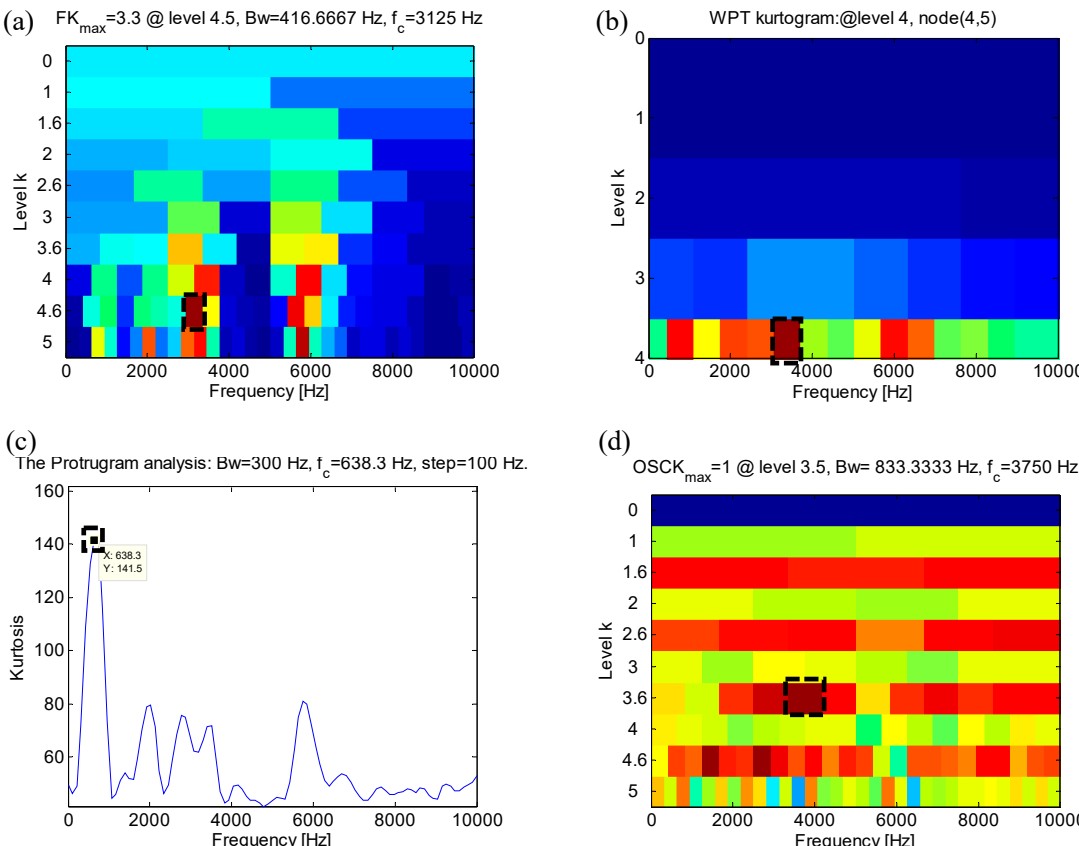

**Figure 22.** The optimal frequency band obtained by different methods: (**a**) FK, (**b**) WPTK, (**c**) Protrugram and (**d**) FOSCK.

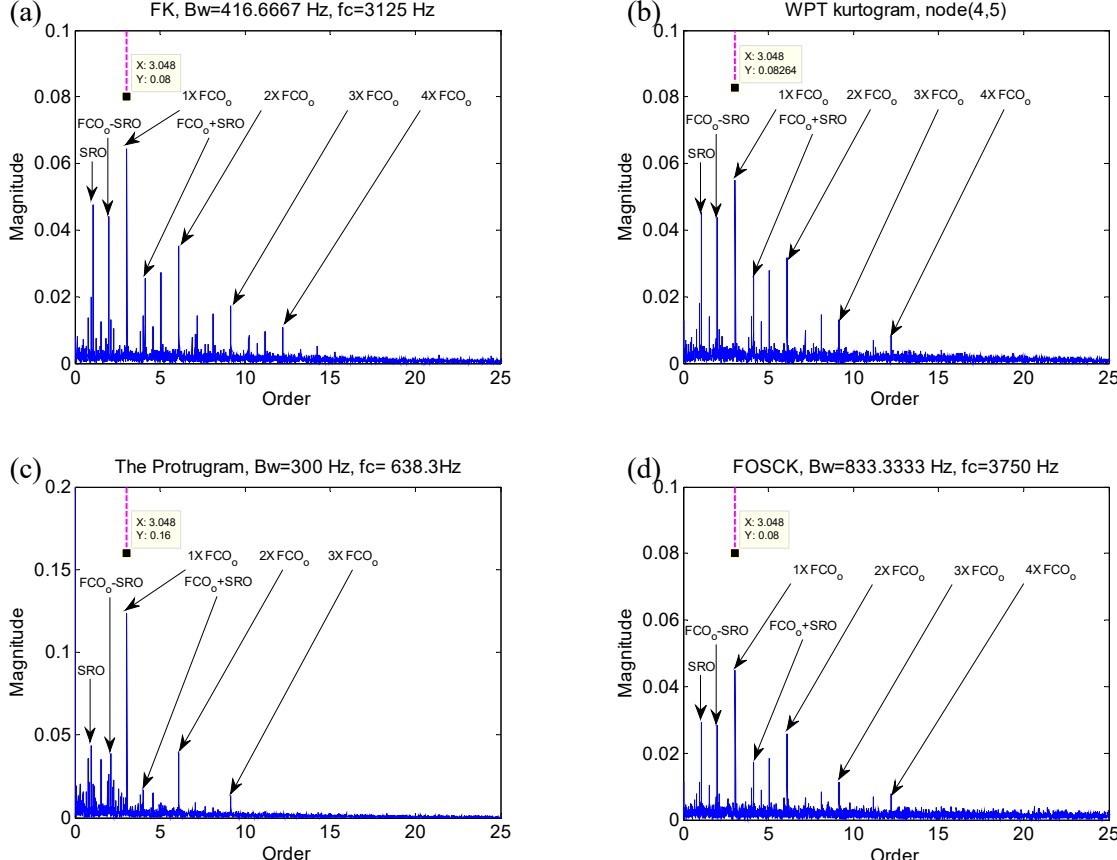

**Figure 23.** The envelope order spectra of the signal obtained by different methods: (**a**) FK, (**b**) WPTK, (**c**) Protrugram and (**d**) FOSCK.

### 4.2.2. Experimental Study #2

To verify that the proposed method is still effective under the conditions of random shock interference, the test is randomly knocked during the changing of speed between 20 Hz and 25 Hz with a equal to 3/2 Hz/s in this experiment, and the collected vibration signal and rotating speed signal are shown in Figure 24a,b, respectively. The acquisition process is disturbed by a transient impact leading to an impulse with a large amplitude in the time domain, which is shown in Figure 24a, and the corresponding frequency spectrum and TFR are shown in Figure 24c,d, respectively. The optimal frequency bands corresponding to different indexes are shown in Figure 25. In Figure 25a,b, the optimal frequency band corresponding to the FK and the WPTK is (4375, 5000) Hz. The Protrugram is paved in Figure 25c with BW = 400 Hz and step = 100 Hz and the center frequency is 744.7 Hz. Figure 25d shows the FOSCK, and its optimal frequency band is (5833.33, 6666.67) Hz. The envelope of the filtered signal from the selected band and its corresponding envelope order spectrum are shown in Figure 26. In Figure 26a,b, they failed to provide any bearing fault related signatures, while in Figure 26c,d, the FCO and its harmonics in the envelope order spectrum can be clearly observed. Therefore, in the case of bearing fault diagnosis under random shock interference, the Protrugram and the FOSCK was better than that of the FK and the WPTK.

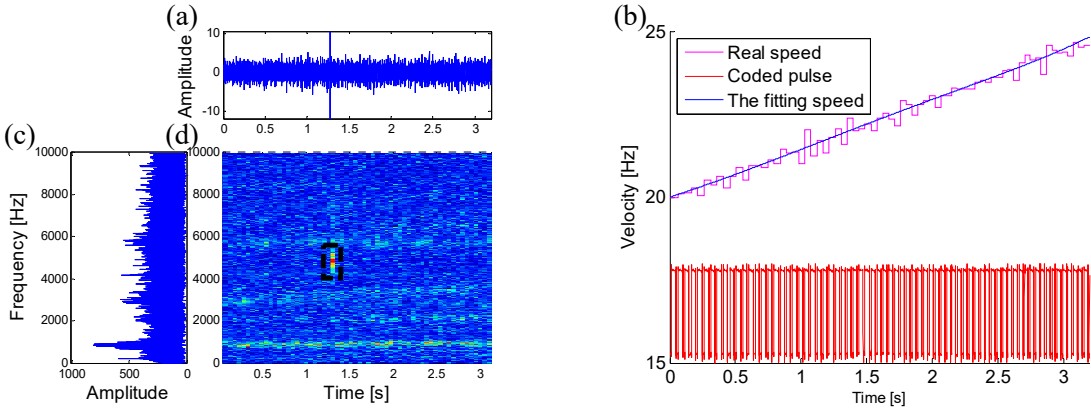

**Figure 24.** The signal measured from an outer race fault bearing: (**a**) time-domain signal, (**b**) the shaft rotational frequency from 20 Hz to 25 Hz, (**c**) the frequency spectrum of (**a**,**d**) TFR by using STFT.

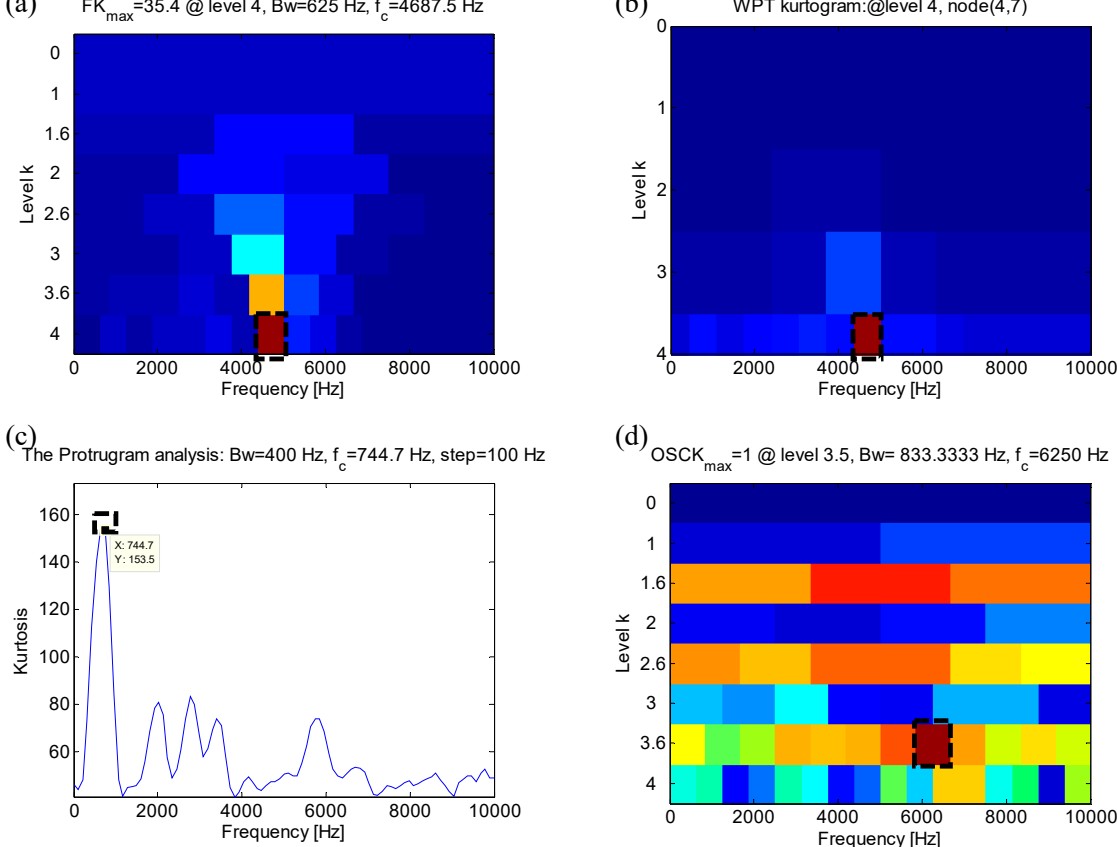

**Figure 25.** The optimal frequency band obtained by different methods: (**a**) FK, (**b**) WPTK, (**c**) Protrugram and (**d**) FOSCK.

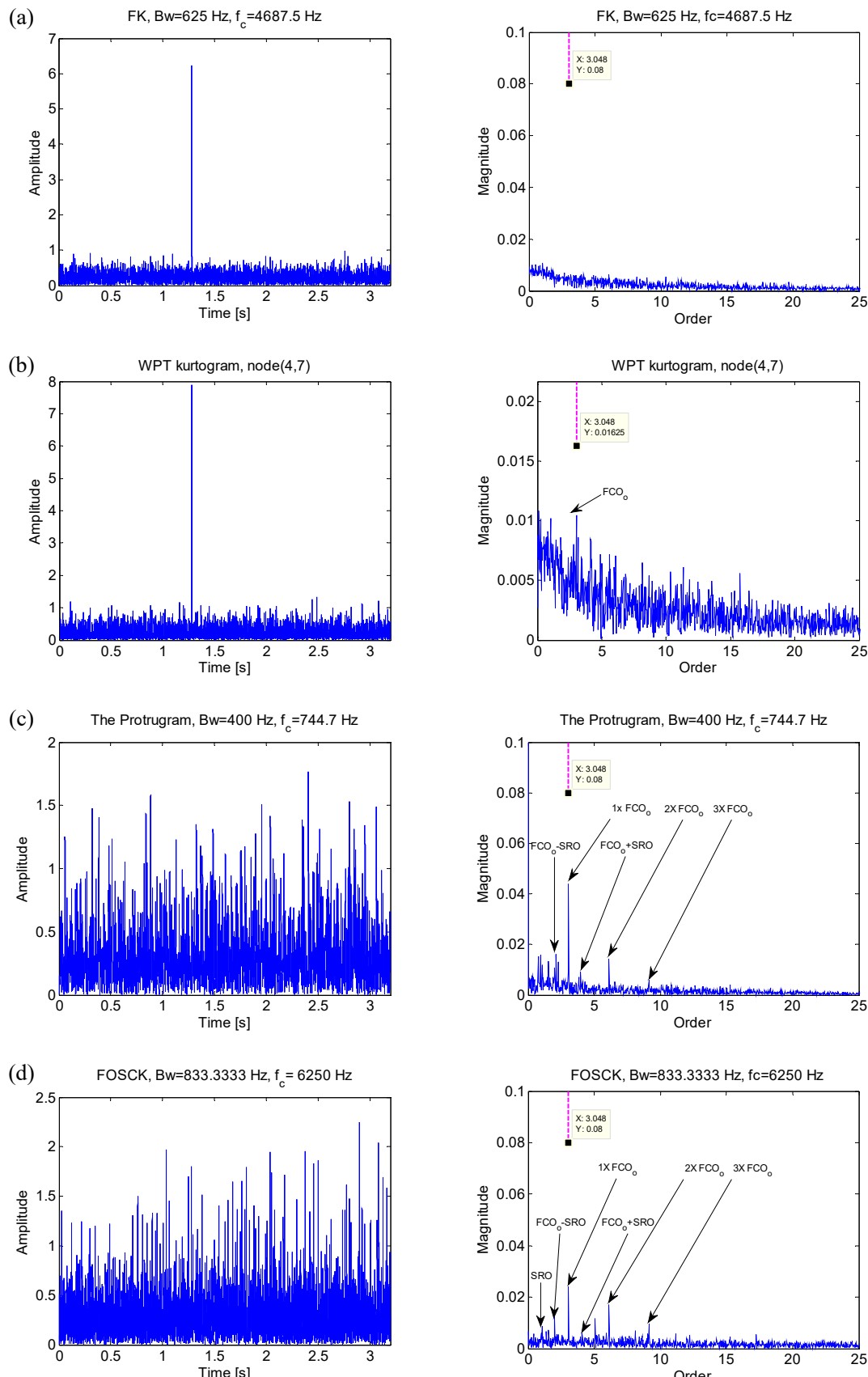

**Figure 26.** The envelope order spectra of the signal obtained by different methods: (**a**) FK, (**b**) WPTK, (**c**) Protrugram and (**d**) FOSCK.

### 4.2.3. Experimental Study #3

In this experiment, to verify that the proposed method is still effective in the case of rapidly changing speed and wild fluctuations, the vibration signal and rotational speed are collected during the speed increase from 18 Hz to 28 Hz with a equal to 3 Hz/s. Figure 27a,b show the raw signal and its rotational speed, respectively. The optimal frequency band corresponding to different indexes is shown in Figure 28. The FK is paved in Figure 28a and the optimal frequency band is [5625, 6250] Hz. The optimal frequency band in the WPTK is [1875, 2500] Hz, as shown in Figure 28b. The Protrugram is shown in Figure 28c, where the BW equals 400 Hz and step is 100 Hz and the center frequency is 2105 Hz. Figure 28d shows the FOSCK, in which the maximum OSCK is calculated at the 3.5th decomposition level, and its corresponding optimal frequency band is [1666.67, 2500] Hz. The envelope order spectrum analysis results are shown in Figure 29a–d. In Figure 29a,d, the FCO and its quadruple octaves are very clear, especially in Figure 29d, and more harmonic components of the FCO can be found. Although the FCO can be found in Figure 29b,c, only the first two octaves are obvious. Therefore, compared with the WPTK and the Protrugram, the FK and the FOSCK are more sensitive to the fault impulse resonance frequency under these conditions.

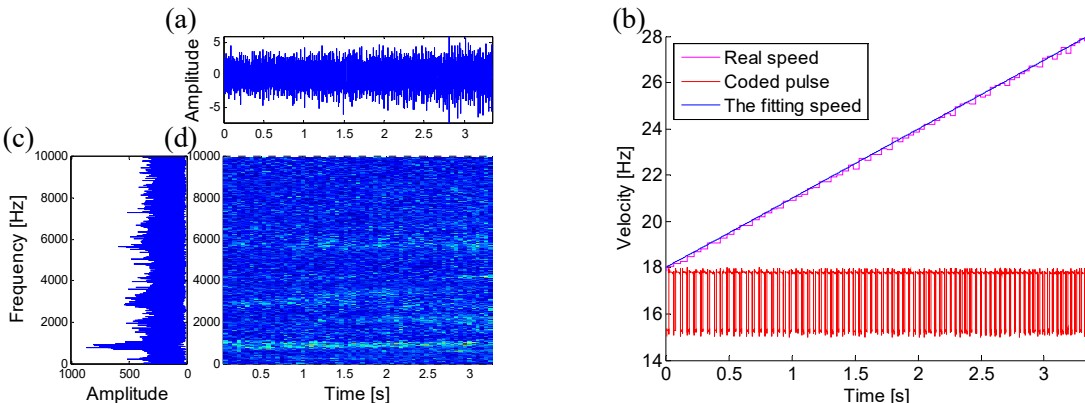

**Figure 27.** The signal measured from an outer race fault bearing: (**a**) time-domain signal, (**b**) the shaft rotational frequency from 18 Hz to 28 Hz, (**c**) the frequency spectrum of (**a**,**d**) TFR by using STFT.

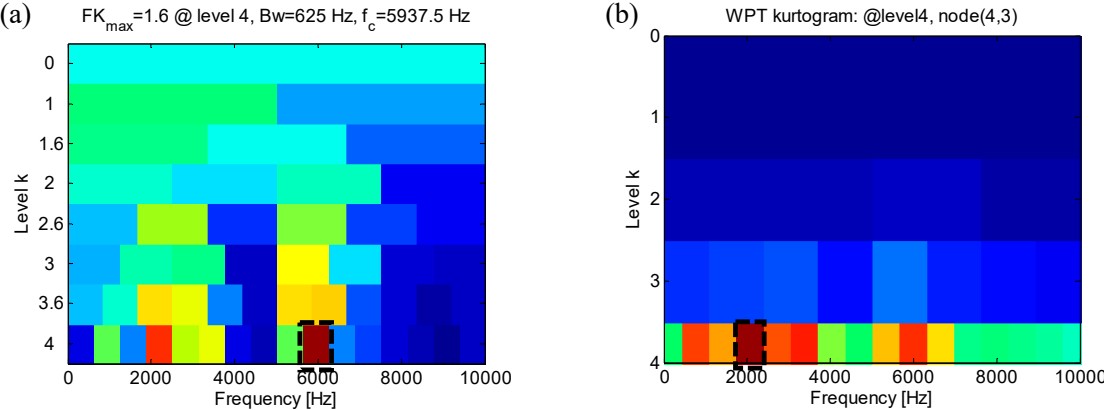

**Figure 28.** *Cont.*

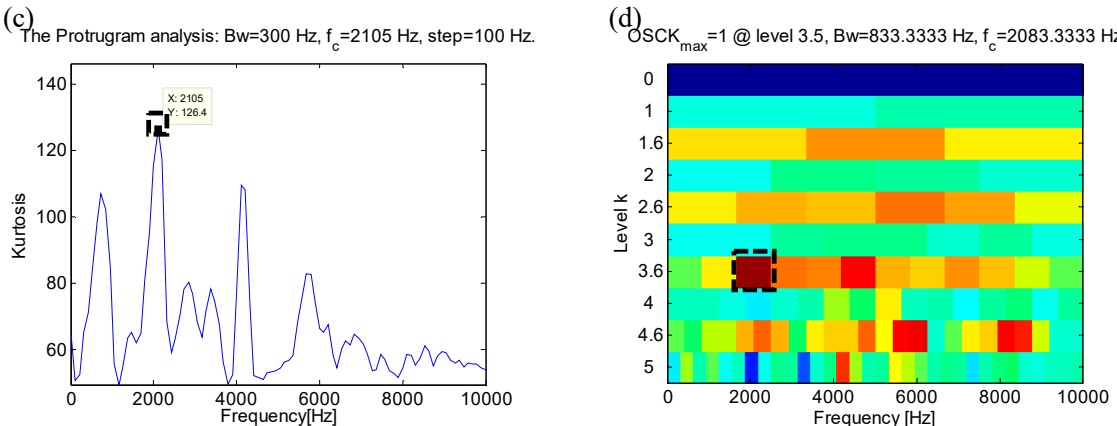

**Figure 28.** The optimal frequency band obtained by different methods: (**a**) FK, (**b**) WPTK, (**c**) Protrugram and (**d**) FOSCK.

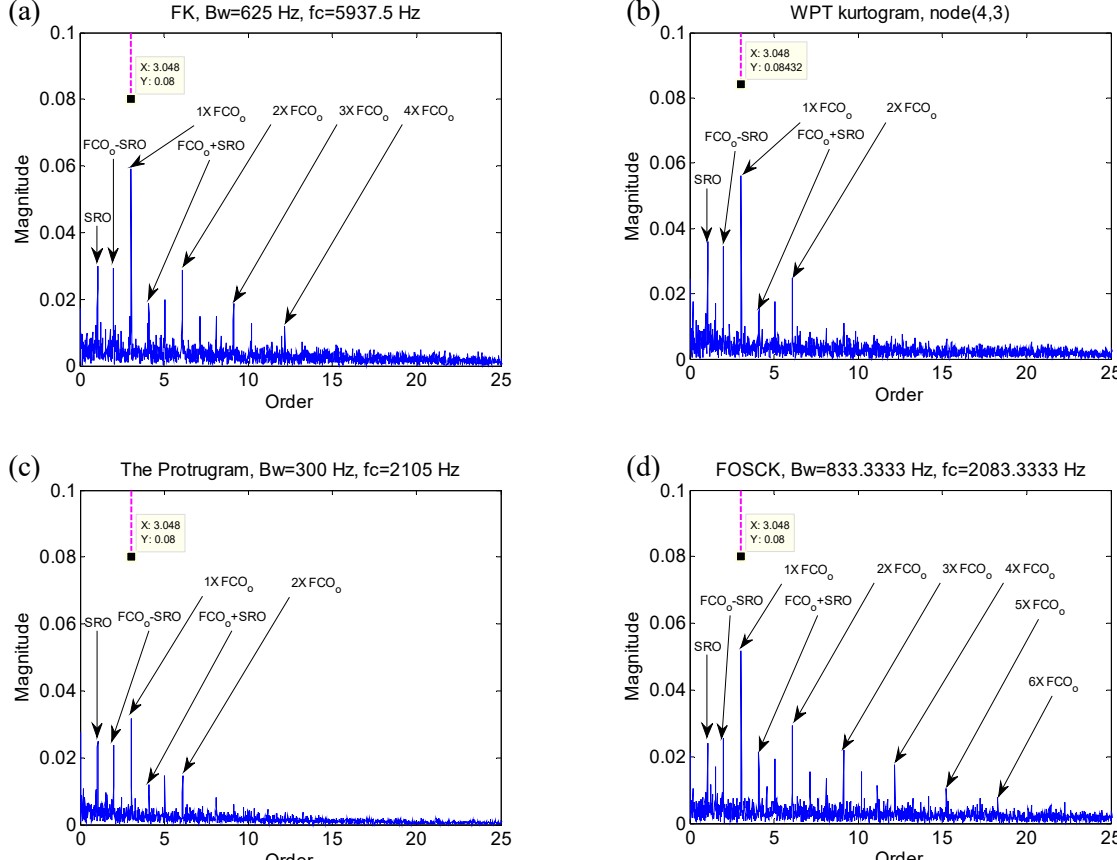

**Figure 29.** The envelope order spectra of the signal obtained by different methods: (**a**) FK, (**b**) WPTK, (**c**) Protrugram and (**d**) FOSCK.

### 4.3. Diagnosis of a Bearing with an Inner Race Fault

In this test, the setup is the same as in the previous test, except that the left normal bearing is replaced by one with an inner race fault, and the right bearing is normal. Three different experiments are carried out to prove the effectiveness of the proposed method for inner race fault diagnosis. White Gaussian noise (SNR=-3 dB) is also added to the collected signal in each experiment. The shaft rotational speed increases from 20 Hz to 25 Hz following a nearly linear pattern with three different accelerations, which are displayed in Table 6.

**Table 6.** Parameters of each experiment.

| Acceleration (Hz/s) | Experimental Study #4 | Experimental Study #5 | Experimental Study #6 |
|---|---|---|---|
| a | 4/3 | 3/2 | 3 |

### 4.3.1. Experimental Study #4

As shown in Figure 30a,b, the vibration signal and the rotating speed are collected at the same time during the speed increase from 20 Hz to 25 Hz, and the acceleration a equals 4/3 Hz/s. The frequency spectrum and TFR of the signal are blurred due to the variable rotating speed and the background noise, as shown in Figure 30c,d.

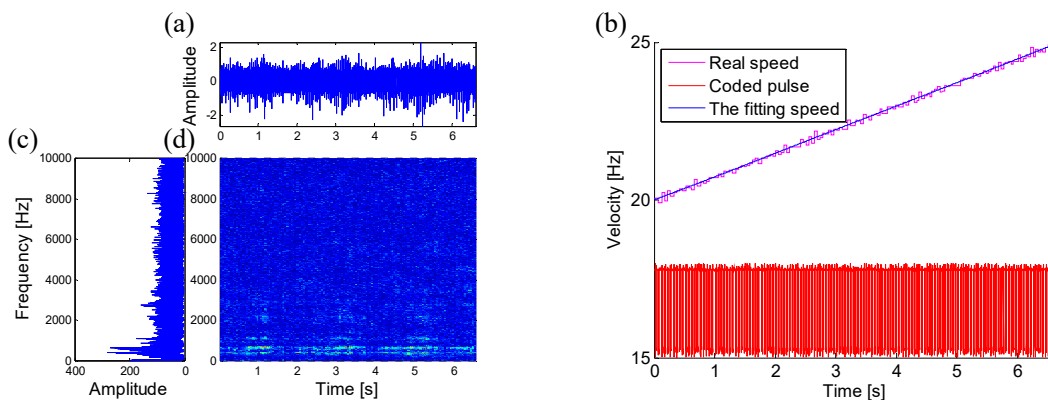

**Figure 30.** The signal measured from an inner race fault bearing: (**a**) time-domain signal, (**b**) the shaft rotational frequency from 20 Hz to 25 Hz, (**c**) the frequency spectrum of (**a**,**d**) TFR by using STFT.

The optimal frequency bands corresponding to different indexes are shown in Figure 31. The FK and the WPTK have the same optimal frequency band, which is [625, 1250] Hz as shown in Figure 31a,b. The Protrugram is paved in Figure 31c with BW equal to 400 Hz and a step of 100 Hz and the center frequency is 425.5 Hz. In Figure 31d, the optimal frequency band of the FOSCK is [3125, 3750] Hz. The envelope order spectra of the filtered signals corresponding to different indexes are displayed in Figure 32. The envelope order spectra do not contain any noticeable FCO in Figure 32a–c, which means that the FK, the WPTK and the Protrugram failed to identify the appropriate fault sensitive resonance frequency band. In Figure 32d, the FCO and its third octaves can be identified in the envelope order spectrum, although the third harmonics are masked by heavy background noise, and the fault component can still be identified. Therefore, the FOSCK has the best ability to detect bearing inner race faults in this case.

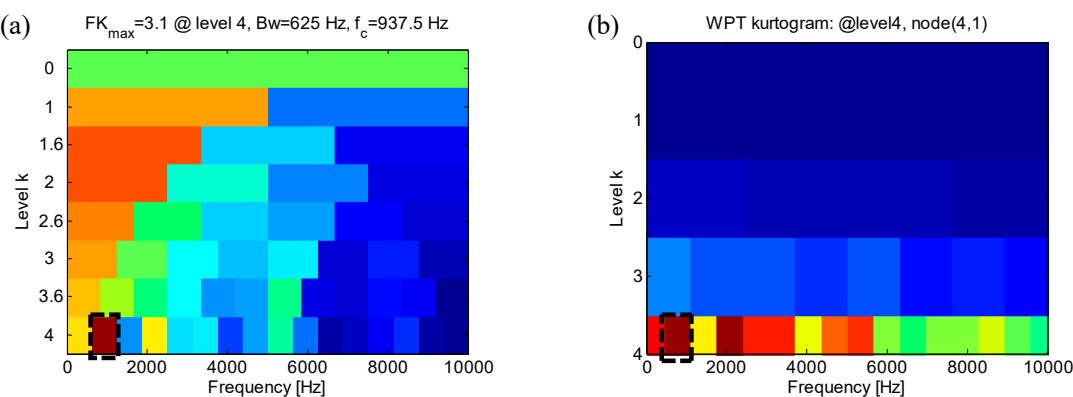

**Figure 31.** *Cont.*

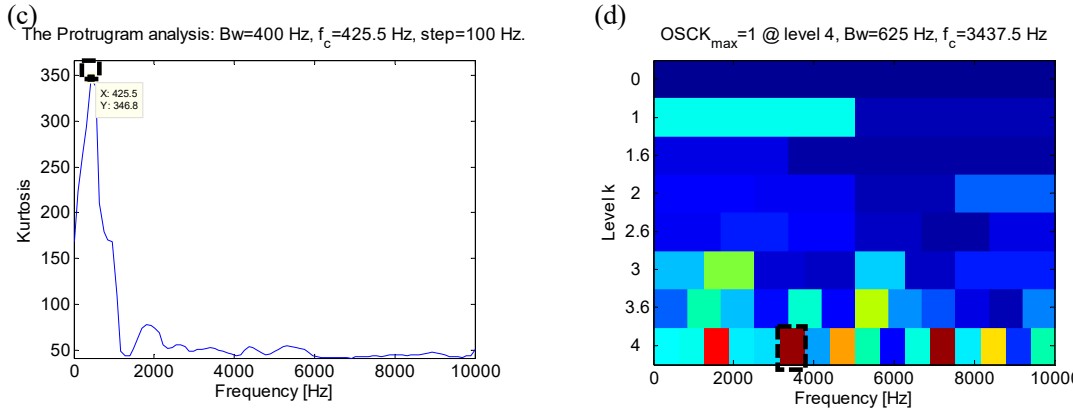

**Figure 31.** The optimal frequency band obtained by different methods: (**a**) FK, (**b**) WPTK, (**c**) Protrugram and (**d**) FOSCK.

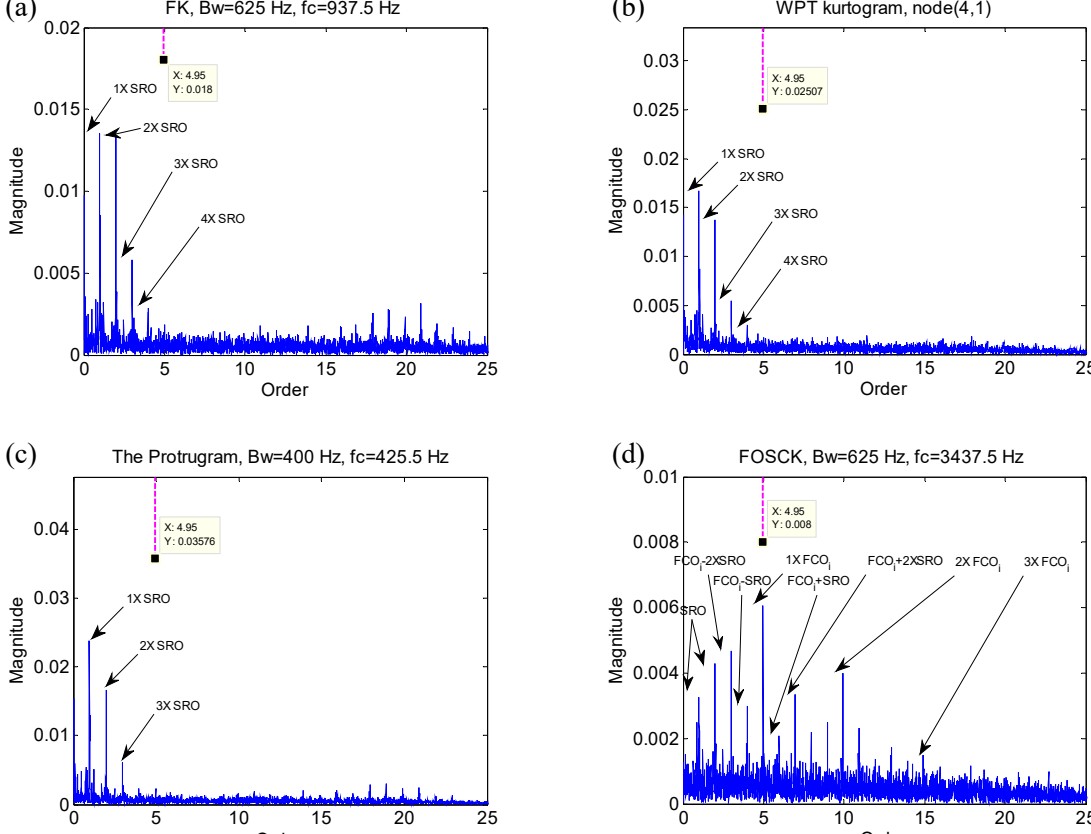

**Figure 32.** The envelope order spectra of the signal obtained by different methods: (**a**) FK, (**b**) WPTK, (**c**) Protrugram and (**d**) FOSCK.

### 4.3.2. Experimental Study #5

In this experiment, the vibration and the rotational speed signal are collected during the speed increase from 20 Hz to 25 Hz with a equal to 3/2 Hz/s, as shown in Figure 33a,b, respectively. As can be seen in Figure 33c,d, the faulty bearing signatures cannot be detected either through the frequency spectrum or the envelope order spectrum directly due to the frequency smearing caused by the varying speed and heavy noise.

The analysis of the vibration signal performed by different methods is shown in Figure 34. One can find in Figure 34a that the maximum kurtosis occurs at level 4, and the optimal frequency band in the FK is [5000, 5625] Hz. The WPTK is shown in Figure 34b, and the optimal frequency band is

[1875, 2500] Hz. Figure 34c shows the Protrugram with BW equal to 400 Hz and step equal to 100 Hz and the center frequency is 537.6 Hz. The optimal frequency band of the FOSCK is [2500, 3750] Hz, as shown in Figure 34d. The results of the envelope order spectrum analysis are shown in Figure 35a–d. The FCO and its third octaves can be identified in the envelope order spectrum obtained by the FOSCK, as shown in Figure 35d. Therefore, the FOSCK has the best ability to detect bearing inner race faults in this case.

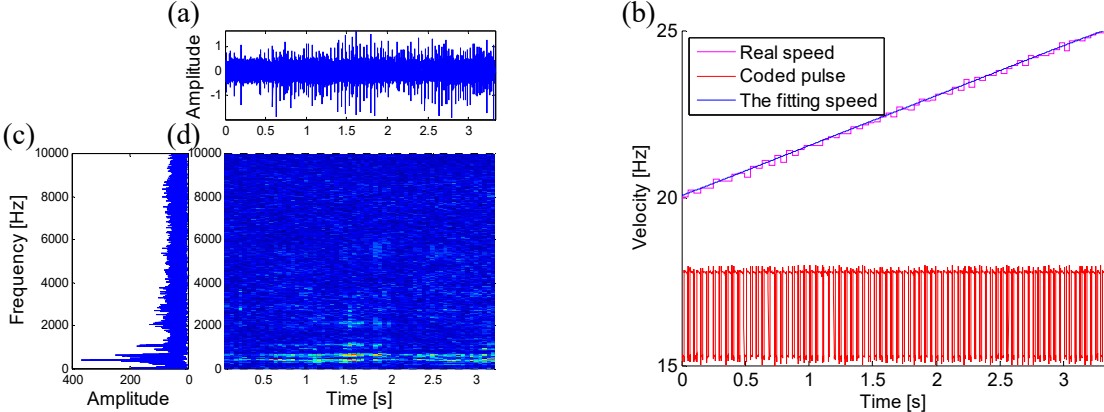

**Figure 33.** The signal measured from an inner race fault bearing: (**a**) time-domain signal, (**b**) the shaft rotational frequency from 20 Hz to 25 Hz, (**c**) the frequency spectrum of (**a**,**d**) TFR by using STFT.

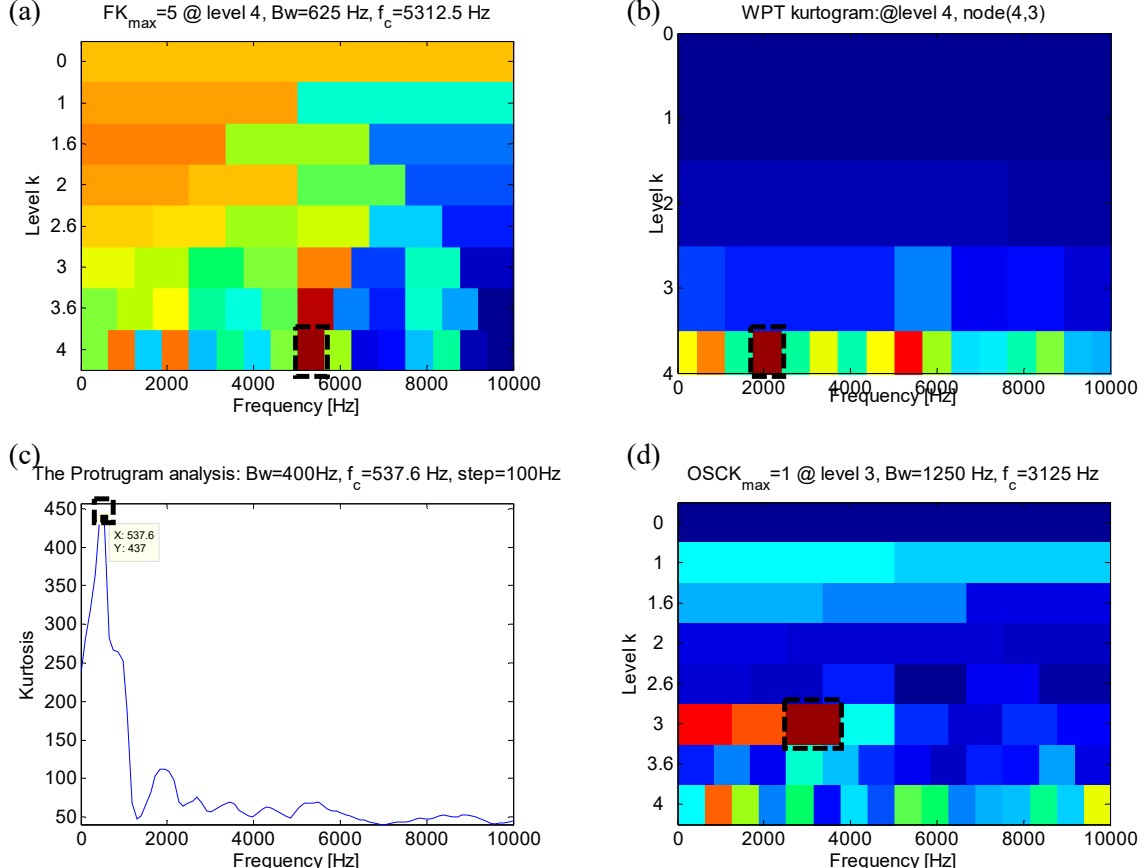

**Figure 34.** The optimal frequency band obtained by different methods: (**a**) FK, (**b**) WPTK, (**c**) Protrugram and (**d**) FOSCK.

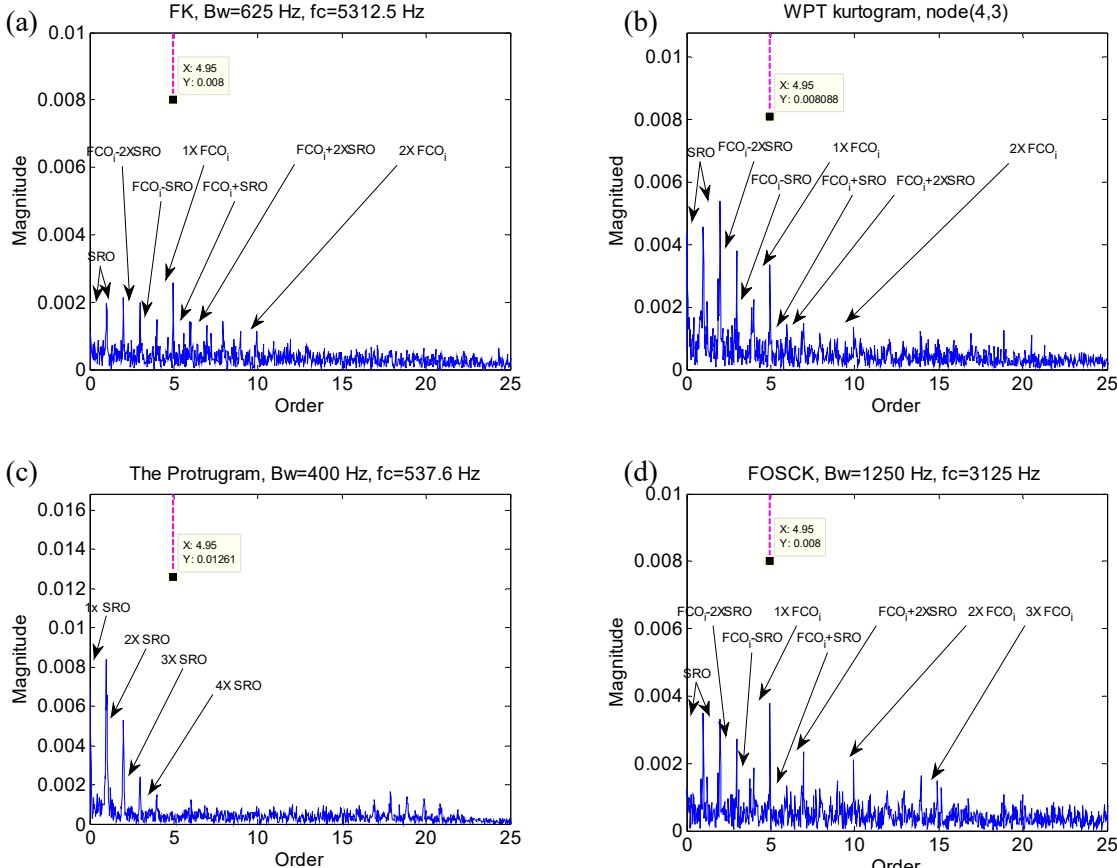

**Figure 35.** The envelope order spectra of the signal obtained by different methods: (**a**) FK, (**b**) WPTK, (**c**) Protrugram and (**d**) FOSCK.

### 4.3.3. Experimental Study #6

Similar to experiment three, the vibration signal and rotational speed are collected as the speed increases from 18 Hz to 28 Hz with a equal to 3 Hz/s to verify that the proposed method is still effective in the case of rapidly changing speed and wild fluctuations. These signals are shown in Figure 36a,b, respectively. The frequency spectrum and the TFR of the vibration signal are shown in Figure 36c,d, respectively.

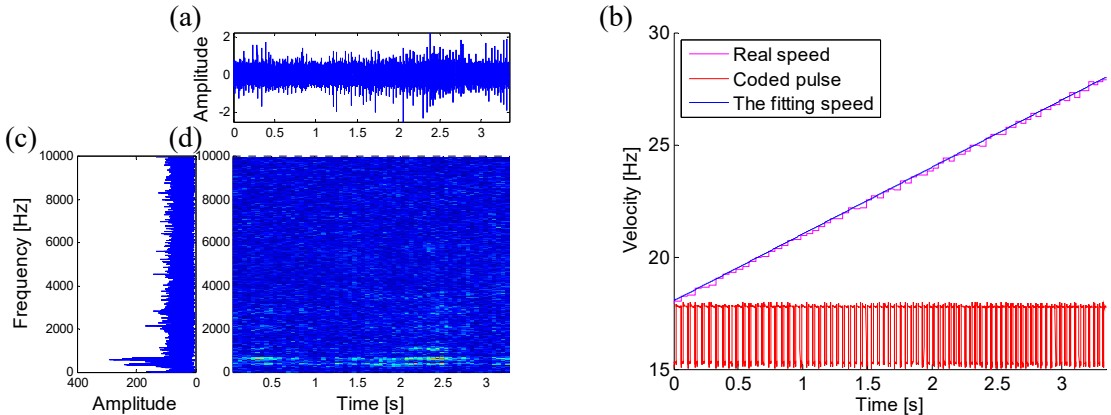

**Figure 36.** The signal measured from an inner race fault bearing: (**a**) time-domain signal, (**b**) the shaft rotational frequency from 18 Hz to 28 Hz, (**c**) the frequency spectrum of (**a**,**d**) TFR by using STFT.

Figure 37 shows the signal analysis results of the inner race fault case. The FK is paved in Figure 37a, in which the optimal frequency band is [625, 1250] Hz. Figure 37b shows the WPTK, in which the maximum kurtosis is calculated at the 4th decomposition level, and its corresponding optimal frequency band is [1875, 2500] Hz. The Protrugram is shown in Figure 37c and the center frequency is 543.5 Hz. Figure 37d shows the FOSCK, and its optimal frequency band is [0, 1250] Hz. The envelope order spectra of the filtered signals are shown in Figure 38a–d. It can be seen that only the envelope order spectrum obtained by the FOSCK can extract the first three octaves in Figure 38d. Therefore, the FOSCK is better than other methods in bearing inner race fault diagnosis in this case.

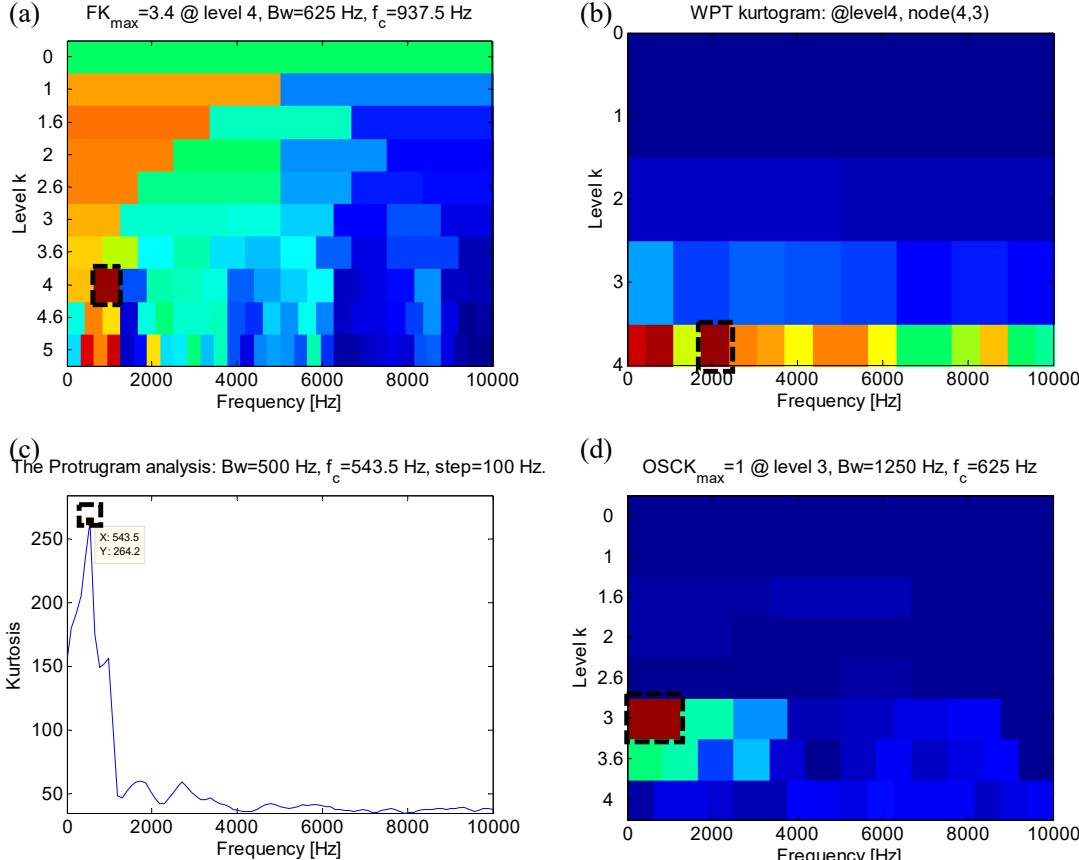

**Figure 37.** The optimal frequency band obtained by different methods: (**a**) FK, (**b**) WPTK, (**c**) Protrugram and (**d**) FOSCK.

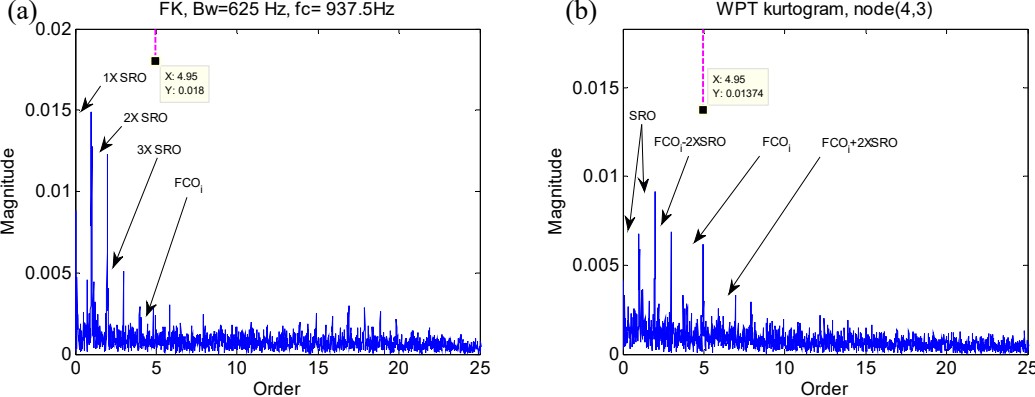

**Figure 38.** *Cont.*

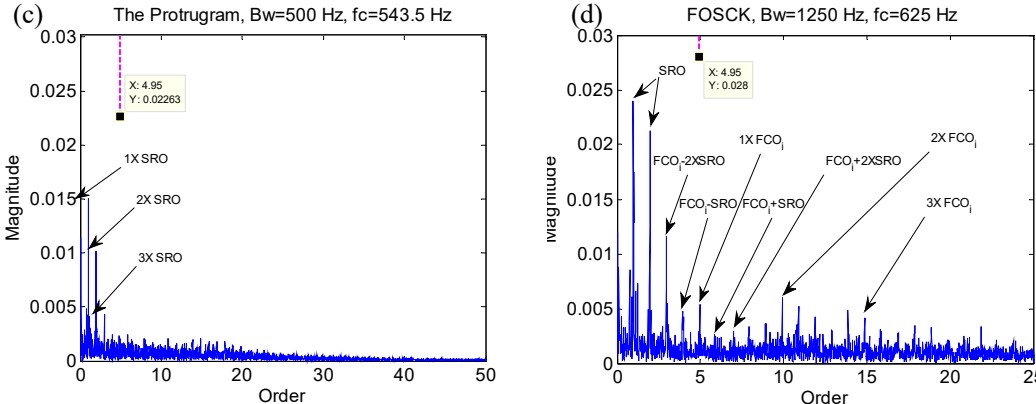

**Figure 38.** The envelope order spectra of the signal obtained by different methods: (**a**) FK, (**b**) WPTK, (**c**) Protrugram and (**d**) FOSCK.

## 5. Discussion

Regarding the various bearing fault signals obtained under different conditions, the diagnoses results are shown in Table 7. Note that the fault is supposed to be diagnosed successfully only if the FCO and its first three harmonics or above are identified effectively, as mentioned in [28].

**Table 7.** Bearing fault diagnosis result.

| Fault Location | Outer Race Fault | | | Inner Race Fault | | |
|---|---|---|---|---|---|---|
| Experimental Study | #1 | #2 | #3 | #4 | #5 | #6 |
| FK | Y | N | Y | N | N | N |
| WPTK | Y | N | N | N | N | N |
| Protrugram | Y | Y | N | N | N | N |
| FOSCK | Y | Y | Y | Y | Y | Y |

\* Y means Yes and N means No.

From the diagnosis results, it is clear that the FOSCK is capable of detecting bearing faults in all cases. It is also concluded that: (1) the fault impact has broadband characteristics and causes different resonances; (2) the Protrugram and FOSCK have better robustness against random impulse disturbance; (3) the FOSCK and FK can diagnose the outer race fault effectively under a large range of the speed fluctuations conditions, which verifies the OSCK index proposed in Section 2 is sensitive to the speed fluctuation while not being affected by the size of the speed; (4) the comparison between the results of experimental studies #1 and #2 shows that both of the Protrugram and FOSCK can suppress the influence of acceleration changes; (5) under the same fault severity conditions, the energy of the inner race fault is dispersed due to the modulation, the local SNR is lower, in which case the envelope order spectra obtained by the existing methods fails to provide any bearing fault related signature. However, the FOSCK is capable of detecting bearing inner race fault in all cases. The results of performance comparison of the FK, the WPTK, the Protrugram and the FOSCK are summarized in Table 8. Besides, considering the FOSCK is robust to the random shock and heavy noise, the method can be applied for exacting random impulses caused by earthquake, in which the random impulses have similar characteristics with bearing fault impulses [29].

**Table 8.** Method robustness.

| Interference | Random Shock | Large Speed Fluctuation | Different Acceleration | Heavy Noise |
|---|---|---|---|---|
| FK | N | Y | P | N |
| WPTK | N | N | P | N |
| Protrugram | Y | N | Y | N |
| FOSCK | Y | Y | Y | Y |

\* Y means Yes, N means No and P means Pending.

## 6. Conclusions

This paper proposes a new feature OSCK based on the COT and CK, and by replacing the OSCK with the kurtosis in the FK, an improved kurtogram the FOSCK is constructed. In the case of simulated signal analysis, the COT procedure may cause warp of the signal resonance band and distortion of the signal amplitude, which means that the COT method must be used after other signal enhancement methods. Compared with other indexes, the OSCK is sensitive to the speed fluctuation while not affected by the size of the speed, so it is more suitable for locating fault-sensitive frequency bands under variable speed conditions. The results of the simulated and experimental bearing vibration signals analyses show that compared with the FK, the WPTK and the Protrugram, the proposed method in this paper can extract fault characteristic information more exactly under different operating conditions and interference environments. In the FOSCK, the COT is carried out many times, which will increases the computational cost. Our work will focus on solving this problem in the future.

**Author Contributions:** Y.R. designed the experiments and analyzed the datasets; W.L., B.Z. and Z.Z. performed the experiments and analyzed part of the dataset; Y.R. and F.J. wrote the paper. All authors contributed to discussing and revising the manuscript.

**Funding:** This work was supported by National Natural Science Foundation of China (No. 51605478), Natural Science Foundation of Jiangsu Province (Nos. BK20160276, BK20160251), China Postdoctoral Science Foundation (No. 2017M621862), Jiangsu Planned Projects for Postdoctoral Research Funds (No.1701193B) and the Project Funded by the Priority Academic Program Development of Jiangsu Higher Education Institutions (PAPD).

**Acknowledgments:** The authors would like to thank all of the reviewers for their constructive comments.

**Conflicts of Interest:** The authors declare no conflict of interest.

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
