# Peer review of "Fault Diagnosis of Rolling Bearings Based on Improved Kurtogram in Varying Speed Conditions"

_applsci, doi:10.3390/app9061157_

Round 1

Reviewer 1 Report

Review report: A number of issues require attention/modification/addition by the authors, before the paper can be recommended for publication. These are: 1- The Abstract should contain a brief sentence, claiming the original contribution(s) made ny the paper. As the authors themselves have noted bearing vibration analysis has been carried out by a plethora of methods and a fair number through use of kurtosis, kurtogram and other variants of the same. So what is the new approach in this paper? 2- The issue of original contribution(s) should be expanded upon in an ultimate paragraph in the introduction, expanding on what is new in this paper with respect to those in open literature. For this purpose the review of literature needs to be expanded to include the following as well, and with respect to these and those already state the authors can claim any original contribution(s). Zhang, Y. and Randall, R.B., “Rolling element bearing fault diagnosis based on the combination of genetic algorithms and fast kurtogram”, Mechanical Systems and Signal Processing, 2009, 23(5), pp. 1509-1517. Wang, L., Liu, Z., Miao, Q. and Zhang, X., “Time–frequency analysis based on ensemble local mean decomposition and fast kurtogram for rotating machinery fault diagnosis”, Mechanical Systems and Signal Processing, 2018, 103, pp. 60-75. 3- There is no need for a paragraph at the end of Introduction, stating what comes next in the paper. This is self-evident. 4- One of the shortcomings of this paper is that it does not really describe the various sources of vibration in bearings which is targeted for recognition and fault attribution by signal processing methods including the kurtogram approaches in this paper. Therefore, there should be some explanation of primary and secondary sources of bearing vibration in the Introduction part of the paper. Only some of these are noted by equations (10)-(12). Please see the papers below, which derive these analytical expressions and provide proper explanations which should be brought to the attention of readers by referring them to these sources: Wardle, F.P., “Vibration forces produced by waviness of the rolling surfaces of thrust loaded ball bearings Part 1: Theory”, Proc. IMechE, Part C: J. Mechanical Engineering Science, 1988, 202(5), pp.305-312. Lynagh, N., Rahnejat, H., Ebrahimi, M. and Aini, R., “Bearing induced vibration in precision high speed routing spindles”, Int. J. Machine Tools and Manufacture, 2000, 40(4), pp. 561-577. Having included the above points, it is essential to note that some sources of bearing vibrations are inherent to its nature and their presence does not constitute a developing fault, such as the variable compliance effect. 5- Another shortcoming is that the paper should also refer to other widely used and established bearing vibration signal processing techniques such as ARMA (Auto-regression moving average method)which does not have the spectral problem of averaging of the signal over the chosen interval as well as side-banding. So, these alternative approaches should have been at least mentioned in the Introduction of the paper and the advantages of the approach described over these noted, if any. Please see below use of ARMA to pin point exact bearing faults such as presence of off-sized rolling elements and cage skidding, etc: Pham, H.T., Yang, B.S., “Estimation and forecasting of machine health condition using ARMA/GARCH model”, Mechanical Systems and Signal Processing, 2010, 24(2), pp. 546-558. Vafaei, S., Rahnejat, H. and Aini, R., “Vibration monitoring of high speed spindles using spectral analysis techniques”, Int. J. Machine Tools and Manufacture, 2002, 42(11), pp. 1223-1234. Vafaei’s use of ARMA to pinpoint an off-sized ball response in a ball bearing is quite similar to the approach to pinpoint a shock in the current paper. So, inclusion of some discussion in the introduction on ARMA is necessary as the alternative approach. 6- Some of the multi-coloured line figures will not be distinguishable in black and white print. Perhaps line types should also be used (re. figure 7(b), for example) 7- I am, by and large, quite happy with the results and discussions. However, please note that the presence of higher harmonics of FCO and such like is often caused by emerging clearances in bearings, which is an assembly/preloading and interference issue and not necessarily an emerging fault. This point should be made when looking at spectra with higher harmonics. Of course the presence of a fault on either races can also cause multiples of cage order. So, how is the method distinguishing between these? 8- In general, I feel that the paper’s result section and discussion can be made more concise. Consider reducing the number of figures. There are really too many. 9- Please remove “[J]” from all references. This is quite unconventional. 10- There should be a full nomenclature of all the mathematical symbols used as well as all the abbreviated terms. I look forward to receiving any reised version, with all the changes made clearly highlighted.

Author Response

Thank you very much for your great efforts in our manuscript. We also appreciate you for your valuable suggestions and questions. According to your advice, we have modified the article. Please refer to the attached word document for details.

Thank you again for everything you have done.

Reviewer 2 Report

See attached.

Author Response

(The authors gave the same response as above.)

Round 2

Reviewer 1 Report

The authors have imple,ented most of my suggestions. The paper is now suitable for publication.

Reviewer 2 Report

Thank you for the editions.